ARTICLES
# Structures of outer-arm dynein array on microtubule doublet reveal a motor coordination mechanism

Qinhui Rao[1,2], Long Han[1,2], Yue Wang[1,2], Pengxin Chai [1,2], Yin-wei Kuo[1], Renbin Yang[1], Fangheng Hu[1], Yuchen Yang[1], Jonathon Howard[1] and Kai Zhang [1✉]

Thousands of outer-arm dyneins (OADs) are arrayed in the axoneme to drive a rhythmic ciliary beat. Coordination among multiple OADs is essential for generating mechanical forces to bend microtubule doublets (MTDs). Using electron microscopy, we determined high-resolution structures of *Tetrahymena thermophila* OAD arrays bound to MTDs in two different states. OAD preferentially binds to MTD protofilaments with a pattern resembling the native tracks for its distinct microtubule-binding domains. Upon MTD binding, free OADs are induced to adopt a stable parallel conformation, primed for array formation. Extensive tail-to-head (TTH) interactions between OADs are observed, which need to be broken for ATP turnover by the dynein motor. We propose that OADs in an array sequentially hydrolyze ATP to slide the MTDs. ATP hydrolysis in turn relaxes the TTH interfaces to effect free nucleotide cycles of downstream OADs. These findings lead to a model explaining how conformational changes in the axoneme produce coordinated action of dyneins.

Eukaryotic cilia and flagella are evolutionarily conserved organelles that are responsible for cellular motility[1], sensory reception[2], embryonic development[3] and intercellular communication[4]. Defects in structures and functions of cilia lead to numerous diseases termed ciliopathies, such as left–right asymmetry in early development, congenital heart defects, hydrocephalus, infertility and primary ciliary dyskinesia (PCD)[5]. Motile cilia are the main drivers for the movement of individual cells and transport of extracellular fluids through periodic ciliary beating[1]. A typical motile cilium is characterized by its '9 + 2' scaffold (Fig. 1a), composed of nine MTDs and a central pair complex (CPC)[6]. Two rows of axonemal dyneins, the OADs and inner-arm dyneins (IADs), power the ciliary beating by sliding the adjacent MTDs. OAD is the key motor protein that generates the majority of mechanical forces required for this fundamental cellular process[7].

A complete OAD is ~1.5–2 megadalton in size and contains two or three (in ciliates and algae) heavy chains (HCs), two intermediate chains (ICs) and a variety of light chains (LCs). Each OAD is divided into two regions, a head that contains AAA+ rings (rings) for ATP hydrolysis and a tail that holds together the whole complex. The tail is permanently attached to the MTD A-tubule, while the head region of each HC contains a microtubule-binding domain (MTBD), which binds and releases the MTD B-tubule depending on the nucleotide states (Fig. 1a)[8–12]. In motile cilia, several thousand OADs are assembled longitudinally along the MTDs as ordered arrays[13–15]. Cryo-electron tomography (cryo-ET) studies suggest that adjacent OADs are indirectly connected to each other via a series of linker structures[13,14]. However, it remains unclear how OAD arrays are formed and why they are important for ciliary beat.

To ensure a rhythmic and energy-efficient beat, OAD molecules need to locally synchronize their conformations and coordinate with each other along the axoneme[16–19]. The coordinated OAD actions are regulated by multiple factors: (1) other axonemal components, such as IADs, nexin-dynein regulatory proteins (N-DRC), radial spokes (RS) and CPC[1]; (2) post-translational modification, such as phosphorylation[20]; (3) extracellular signals, for example, $Ca^{2+}$ concentration[21] and redox states[22]; and (4) local curvatures of the axoneme[23]. In some cases, cilia without IAD, RS or CPC are also capable of producing a rhythmic beat, implying the existence of a dedicated sensory system among OAD molecules themselves to accomplish motor coordination[24]. However, the mechanism underlying motor coordination remains elusive. Here, using cryo-EM, we set out to study how conformations of OAD are correlated to its microtubule-binding state (MTBS) alteration and how conformational changes of each OAD unit along the array affect its downstream neighbors.

We use the model system *Tetrahymena thermophila* for biochemical and cryo-EM analysis in this study. Its OAD contains three HCs (α-, β- and γ-HC), two ICs (IC2 and IC3) and a set of indefinite LCs. We reconstituted the purified OADs onto MTDs to mimic native arrays and determined the structures, by cryo-EM, of free OAD and OAD arrays bound to MTDs (OAD-MTD) in two different microtubule-binding states. We show that microtubule binding induces free OADs to spontaneously adopt a parallel conformation, which is primed for array formation in a TTH manner. The conformations of arrayed OADs are synchronized in either microtubule-binding state. The array involves an extensive network of interactions and is coordinately remodeled when OADs take one step forward. The TTH interactions remain nearly unchanged in both states, but need to be broken to allow MTBS alteration during the mechanochemical cycle. Nucleotide treatment on the OAD-MTD array reveals that the TTH interactions between neighboring OADs gate the ATP hydrolysis of downstream motors. Opening the gate requires the release of their upstream neighbors from microtubules by ATP hydrolysis. In combination with previously reported cryo-ET structures, we propose a model for how OADs coordinate with each other to move one step on the MTD, and how ciliary beat is propagated.

¹Department of Molecular Biophysics and Biochemistry, Yale University, New Haven, CT, USA. ²These authors contributed equally: Qinhui Rao, Long Han, Yue Wang, Pengxin Chai. ✉e-mail: jack.zhang@yale.edu

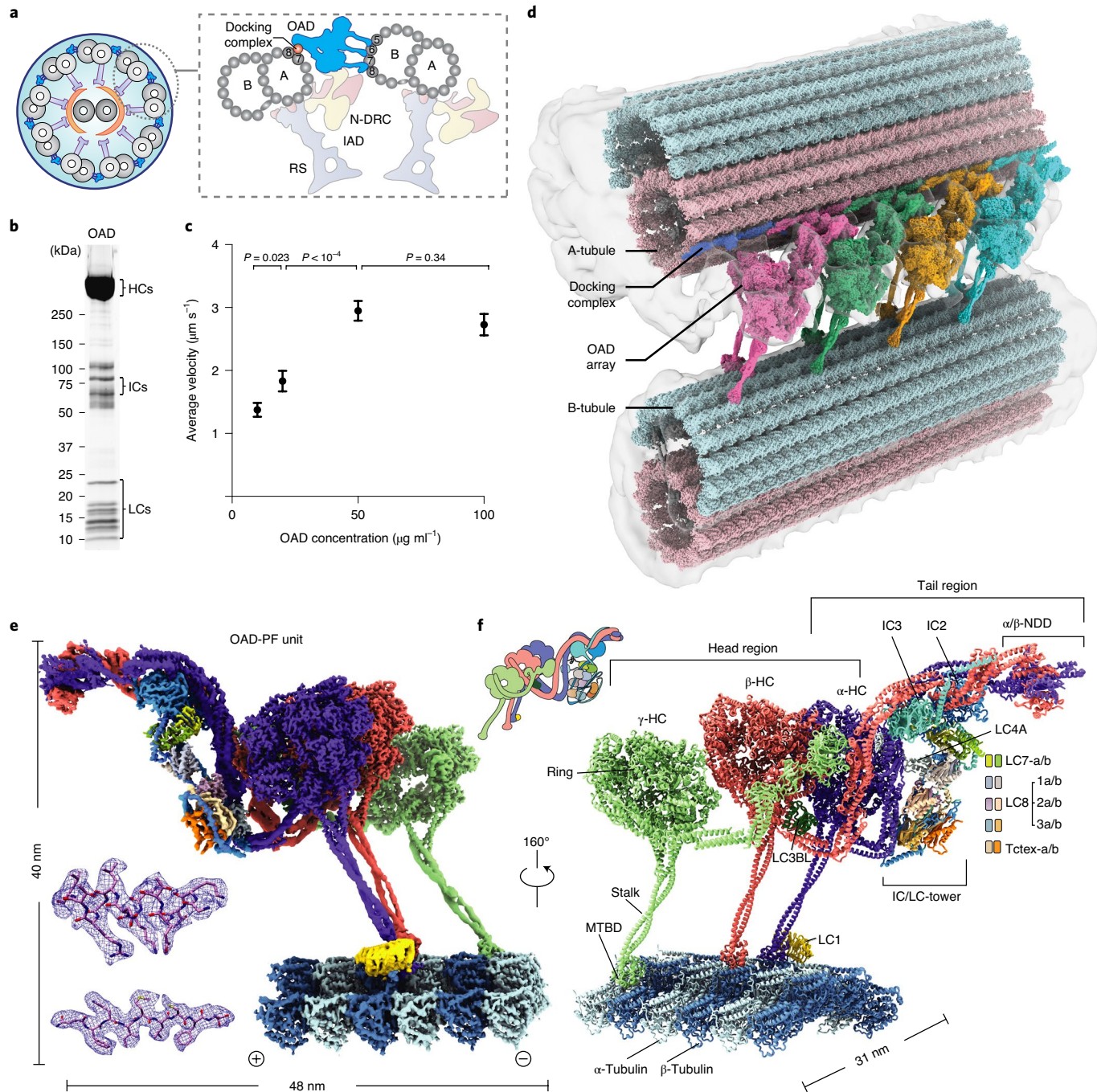

**Fig. 1 | Overall structure of OAD array bound to MTD. a**, Diagrams of the *T. thermophila* cilium and OAD bound to two adjacent MTDs. **b**, Representative SDS–PAGE of free OAD stained by SYPRO Ruby. OAD purifications were reproduced more than five times. **c**, The mean velocities of microtubule gliding measured by using a series of free OAD concentrations at 10 µg ml$^{-1}$ (*n* = 78 microtubules), 20 µg ml$^{-1}$ (*n* = 65), 50 µg ml$^{-1}$ (*n* = 75) and 100 µg ml$^{-1}$ (*n* = 51) from three replicates. Data are presented as mean values ± s.d. and *P* values were calculated using Welch's *t*-test. **d**, Side view of the cryo-EM structure of the OAD array bound to two adjacent MTDs. The map is generated by fitting single-particle cryo-EM maps of the OAD array and MTDs into our cryo-ET map and a previously reported reconstruction (EMD-7805)[9]. The docking complex model was based on a *C. reinhardtii* homolog (PDB 6U42)[15]. **e**, Density map of one OAD-PF unit in MTBS-1. The map is a combination of 31 locally refined regions. **f**, A molecular model of the OAD-PF structure built from the map in **e**. Suffixes of LC7, LC8 and Tctex are based on their relative positions, 'b' for those relative to the microtubule minus end and 'a' for the opposite position. Source data for **b** are available online.

## Results

**Two microtubule-binding states of parallel OAD arrays.** Native OAD and MTD were extracted from the isolated cilia of *T. thermophila*. After demembranation and high salt treatment on the purified axoneme, the MTD was enriched by centrifugation and OAD was purified by sucrose-gradient ultracentrifugation followed by ion exchange for all subsequent biochemical and structural analyses (Fig. 1b,c and Extended Data Fig. 1). To assemble the OAD-MTD complex in vitro, we incubated the freshly purified native OAD (not cross-linked) and MTD in the reconstitution buffer

**Table 1 | Cryo-EM data collection, refinement and validation statistics**

| | OAD-MTD array (apo state) | | | Free OAD (apo state) |
|---|---|---|---|---|
| **Data collection and processing** | Dataset 1 | Dataset 2 | Dataset 3 | Dataset 4 |
| Magnification | ×165,000 | ×165,000 | ×105,000 | ×105,000 |
| Voltage (kV) | 300 | 300 | 300 | 300 |
| Electron exposure ($e^-$/$Å^2$) | 53.3 | 53.3 | 53.3 | 53.3 |
| Defocus range (μm) | −1.0 to −2.0 | −1.0 to −2.0 | −1.5 to −2.5 | −1.5 to −2.5 |
| Pixel size (Å) | 0.822 | 0.822 | 1.333 | 1.333 |
| Symmetry imposed | $C1$ | $C1$ | $C1$ | $C1$ |
| Initial particle images (no.) | 207,795 | 616,864 | 2,022,385 | 883,738 |
| Final particle images (no.) | 6,895 | 7,288 | 188,989 | 209,656 |
| Merged particles images (no.) | 268,712 | | | 209,656 |
| Final particles for refinement (no.) | 191,776 | 76,936 | | 209,656 |
| | OAD-PF MTBS-1 (EMD-22677) (PDB 7K58) | OAD-PF MTBS-2 (EMD-22679) (PDB 7K5B) | | Preparallel OAD (EMD-22840) (PDB 7KEK) |
| Map resolution (Å) | 3.5 | 4.5 | | 8.0 |
| FSC threshold | 0.143 | 0.143 | | 0.143 |
| Map resolution range (Å) | 2.8-10 | 3.8-6.0 | | 5.0-12 |
| **Refinement** | | | | |
| Initial model used | Ab initio | Ab initio | Ab initio | Ab initio |
| Model resolution (Å) | 3.5 | 3.96 | 3.82 | 3.83 |
| FSC threshold | 0.143 | 0.143 | 0.143 | 0.143 |
| Model resolution range (Å) | 3.5-7.0 | | | |
| Map sharpening $B$ factor ($Å^2$) | −80 | −80 | −80 | −80 |
| Model composition | Entire | α-Motor | β-Motor | γ-Motor |
| Nonhydrogen atoms | 201,509 | 21,493 | 21,535 | 23,540 |
| Protein residues | 26,111 | 2,671 | 2,661 | 2,924 |
| Ligands | ATP/ADP/ $Mg^{2+}$ (3/6/9); | ATP/ADP/ $Mg^{2+}$ (1/2/3) | ATP/ADP/ $Mg^{2+}$ (1/2/3) | ATP/ADP/ $Mg^{2+}$ (1/2/3) |
| | GTP/GDP/$Mg^{2+}$ (12/12/12) | | | |
| **$B$ factors ($Å^2$)** | | | | |
| Protein | 145.912 | 207.098 | 214.995 | 230.879 |
| Ligand | | | | |
| **R.m.s. deviations** | | | | |
| Bond lengths (Å) | 0.0030 | 0.0031 | 0.0027 | 0.0027 |
| Bond angles (°) | 1.3386 | 1.3286 | 1.2985 | 1.3010 |
| **Validation** | | | | |
| MolProbity score | 2.00 (100th) | 1.90 (100th) | 1.87 (100th) | 1.71 (100th) |
| Clashscore | 4.68 (100th) | 4.56 (100th) | 4.75 (100th) | 3.92 (100th) |
| Poor rotamers (%) | 2.21 | 2.57 | 3.92 | 2.58 |
| **Ramachandran plot** | | | | |
| Favored (%) | 91.78 | 94.51 | 95.54 | 94.09 |
| Allowed (%) | 7.31 | 5.08 | 4.20 | 5.36 |
| Disallowed (%) | 0.91 | 0.41 | 0.26 | 0.55 |

on ice and optimized the molar ratio by negative-stain screening (Extended Data Fig. 1c). Microtubule-gliding assays indicated that the isolated OADs are active in vitro (Supplementary Video 1). The gliding velocity is positively correlated with the OAD concentration as well as with the microtubule length (Fig. 1c and Extended Data Fig. 1a,b)[25,26]. Early evidence showed that the microtubule sliding is strikingly fast if OADs are aligned in bundles[27]. Studies also revealed that the sliding could be stopped by blockage of several adjacent

OADs, suggesting that OADs are successively activated along an MTD[19]. In addition, our in-vitro reconstitution assay indicates that ordered OAD arrays can be spontaneously formed in the presence of MTDs (Extended Data Fig. 1c), in line with previous findings[16]. These findings together suggest that multiple OADs tend to unify their forces to slide microtubules. Therefore, direct coordination between OADs is very likely to occur during beating.

To elucidate the structural basis for how OADs coordinate their actions, we performed cryo-EM analysis on the reconstituted OAD-MTD arrays (Extended Data Figs. 1d–f, 2 and 3 and Table 1). Two-dimensional (2D) analysis indicates that the reconstituted OAD-MTD is mediated by the MTBDs rather than the tail of OADs in the absence of docking complex. Cryo-EM classification reveals that the OAD arrays adopt two distinct microtubule-binding states, MTBS-1 and MTBS-2 (Extended Data Fig. 1e,f). In both states, the relative axial positions for α- and γ-MTBD are fixed on MTD (γ-MTBD is always 8 nm ahead), while the β-MTBD position is equivalent to either α-MTBD in MTBS-1 or γ-MTBD in MTBS-2 (Extended Data Fig. 1e,f). The OAD unit together with the four binding protofilaments (OAD-PF) was locally refined to resolutions of ~2.8–3.8 Å for most structured regions in MTBS-1 and to resolutions of ~3.8–6 Å in MTBS-2 (Fig. 1d,e, Extended Data Fig. 3 and Supplementary Video 2). By combining mass spectrometry and a genome-wide pattern search, 18 unique subunits of OAD, including three HCs (α-, β- and γ-HC), two ICs (IC2 and IC3) and 13 different LCs (LC7-a/b, LC8-1a/b, LC8-2a/b, LC8-3a/b, Tctex-a/b, LC4A, LC3BL and LC1), were unambiguously identified, built and refined (Fig. 1f and Supplementary Table 1).

The head of each HC comprises an AAA+ ring (AAA1–AAA6) for ATP hydrolysis, a six-helix (H1–H6) MTBD, which binds microtubule, a coiled-coil stalk (helices CC1 and CC2), which connects the ring to MTBD, and a linker (Extended Data Fig. 4a), which links the ring to the tail (Fig. 1f)[28]. The tail region of α-HC and β-HC (Extended Data Fig. 4b) has a similar topology to that of cytoplasmic dyneins and consists of nine helical bundles (HB1–HB9)[25], while γ-HC has a special tail (Extended Data Fig. 4c). The arrayed OADs adopt a fully parallel conformation in which the three AAA+ rings are stacked together via linker–ring (LR) interactions (Fig. 1d,e and Extended Data Fig. 4d). The OAD arrays fit well into previously reported in-situ cryo-ET maps (Extended Data Fig. 4e)[9–11,17], indicating that our reconstituted OAD arrays reflect the native structures in cilia in the apo state (Extended Data Fig. 4f). Our structure shows that α-HC, β-HC, IC2 and IC3 form the core structure of OAD (core-OAD) (Fig. 1f). The core-OAD is conserved across species and all the four subunits are critical for OAD assembly[1]. Together they serve as a scaffold for binding all other chains and mediate nearly all interactions between two adjacent OADs.

**The structure of γ-HC and its special linker docking mode.** Cryo-EM reconstruction reveals that the γ-tail comprises a six-bladed kelch domain (γ-kelch), followed by two immunoglobulin folds (Ig-PT and Ig-Fln)[1,29] (Extended Data Fig. 4c). The special γ-HC (Fig. 2a,b) and ten IC-bound light chains (Fig. 2c,d) flank the core-OAD and play essential roles in regulating OAD activity[8,30–34]. γ-HC extends out from the core-OAD array with its γ-kelch tightly bound to HB6 of the β-tail (β-HB6) via a snug insertion of two consecutive helical segments (residues P952–T973) into the γ-kelch

groove (Extended Data Fig. 5a). The following Ig-fold region joins to the neck region (equivalent to HB8–HB9 of α- or β-HC), which is further connected to the γ-ring by γ-linker (Fig. 2a). The γ-ring is pinched between its own linker and the adjacent β-linker (Extended Data Fig. 5b).

Compared to the unprimed cytoplasmic dyneins, or α- and β-HC of OAD in MTBS-1, which all adopt the classical post-powerstroke state[35], defined as post-1, γ-HC adopts a distinct powerstroke state, defined as post-2, in which its linker is upraised toward AAA3–AAA4 (Fig. 2b). This leads to a relative slide between the γ-linker subdomain 0/1 (γ-LS0/1) and helix H2 in the PS-I region of the γ-AAA5 large subunit (AAA5L). The interaction interfaces between γ-linker and AAA5L are thus changed, and two additional docking sites on the γ-ring are introduced. One is the γ-AAA2L H2 insert, which contacts the start of the helix H7 in γ-LS1 (Fig. 2b). The other site is the AAA3 extension (AAA3E), which interacts with γ-LS0 (Fig. 2b).

**Structures of LCs and their roles in regulating OAD activity.** Among all the 13 light chains, 10 of them (LC7, LC8 and Tctex families) are clustered by the IC2/3 N-terminal extensions (NTEs) (Extended Data Fig. 5c). Together they form a tower-like structure (IC/LC-tower) (Fig. 2c) and attach to the α-tail en bloc (Fig. 1f). Each LC occupies a unique position surrounded by other LCs with specific interfaces (Fig. 2c,d and Extended Data Fig. 5d). Compared to cytoplasmic dyneins, which have linear organizations of homodimeric light chains, OAD has a heterodimeric Tctex, which is folded back and pinned to LC8s, attributed to a special helical bar (residues N121–E147) and a β-hairpin (residues N85–E100) of IC2 (Fig. 2c). We fitted our OAD structure into a previously reported cryo-ET structure of axoneme, which shows that the back-folded Tctex region links the IC/LC-tower to either N-DRC or IADf (Extended Data Fig. 5e). With more than ten different proteins interwoven, this local region is tightly attached to the HB8 of the α-neck and is likely to mediate the communication between N-DRC (or IADf) and OAD arrays.

The other three LCs, including an LC3B-like subunit (LC3BL), LC4A and LC1, are separately located in different regions of the OAD. LC3BL belongs to the thioredoxin family and is involved in redox-based control of OAD activities[33]. It binds to the joint region between β-LS0 and β-HB9 and contacts γ-Ig-PT, linking the γ-tail to the β-linker (Extended Data Fig. 5b). These multidomain contacts together form a local network and potentially regulate the activity of β- and γ-HC. The calmodulin LC4A regulates OAD activity in a calcium-dependent manner[32]. In the current OAD structure, it adopts a typical calcium-free state, binds to α-HB6 and links the α-tail to the IC/LC-tower (Extended Data Fig. 5f). LC1 is the only known dynein light chain that binds to α-MTBD for OAD activity regulation, and was previously proposed to be a potential mechanical sensor for curvature response during ciliary beating[8,34] (Fig. 1e,f).

**MTBDs recognize their native-like tracks collectively.** Different from cytoplasmic dynein-1, which requires dynactin to align the two motor domains for activation[25], OADs spontaneously form a parallel conformation upon microtubule binding (Extended Data Fig. 1c,d). OAD units along each array are locally synchronized (Extended Data Fig. 6a). However, the inter-PF angles (defined as

**Fig. 2 | Structures of the γ-HC and IC/LC-tower. a,** The overall architecture of γ-HC, which has a special tail and is distinct from the classical heavy chain. **b,** The γ-linker docks on the γ-ring in a new mode (post-2), which has three distinct linker–ring interaction sites indicated by colored pentagrams (blue, AAA2L H2 insert/LS1; green, AAA5 PS-I/LS0-1; red, AAA3E/LS0). **c,** The overall architecture of the IC/LC-tower. The tower consists of two ICs and five heterodimers of LCs, the IC2/3, LC7-a/b, LC8-1a/b, LC8-2a/b, LC8-3a/b and Tctex-a/b. In addition to the two cross-overs and five contact sites between IC2/3 NTEs, a helical bar (residues N121–E147) and a β-hairpin (residues N85–E100) of IC2 force the Tctex-a/b dimer to fold back and attach to the side of LC8-2a/b and LC8-3a/b, facing toward the inner side of the OAD array. **d,** A schematic of the interaction interface between ICs and LCs.

the relative rotation between two adjacent protofilaments[36]) vary substantially among all pairs of MTD protofilaments (Extended Data Fig. 6b,c). To preserve a stable parallel architecture, the OAD stalks need to rotate with respect to the MTBD regions to compensate for the inter-PF angles. This is enabled by the hinges between stalk CC1 and the MTBD helix H1 (Fig. 3a). We compared the

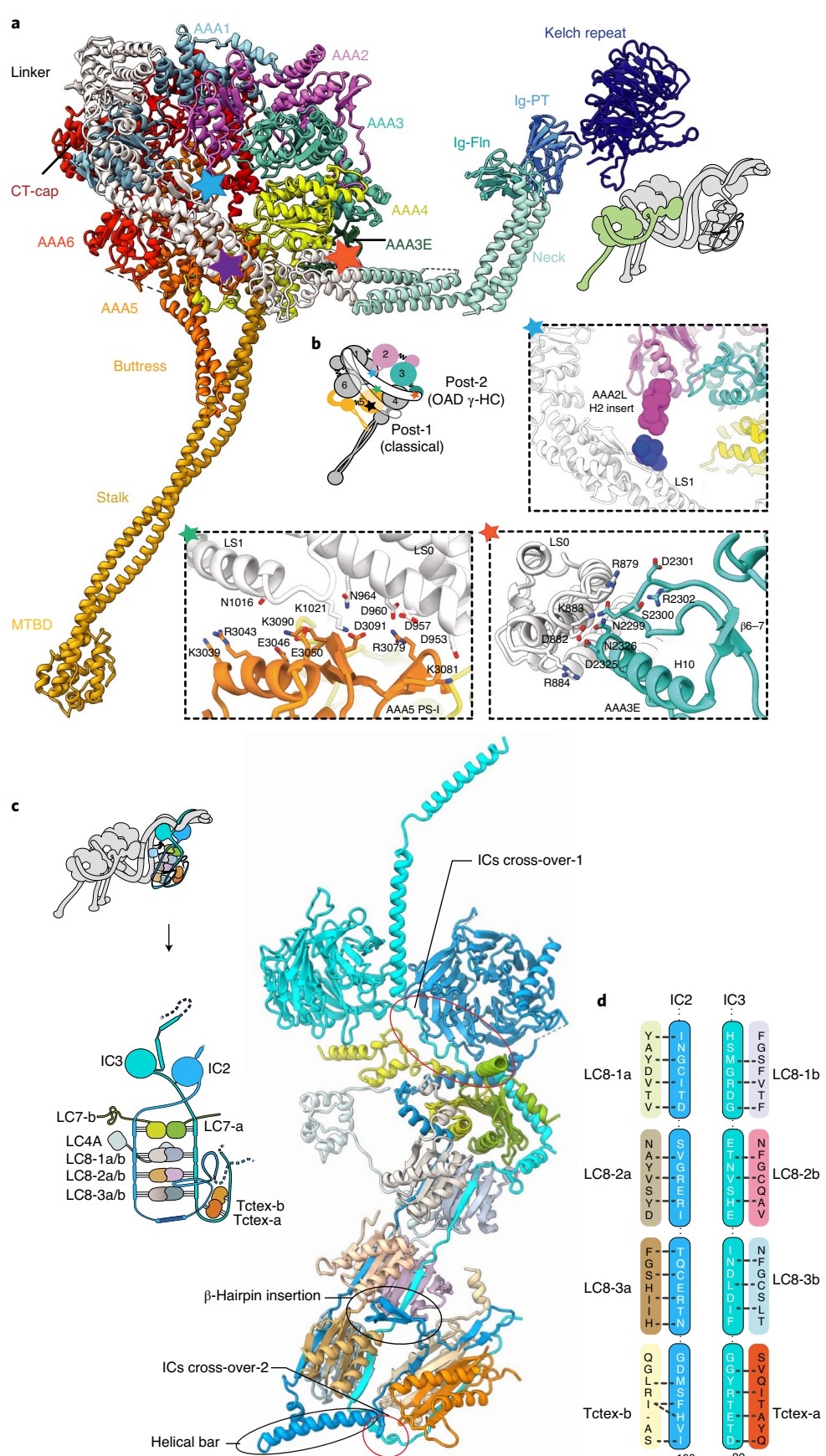

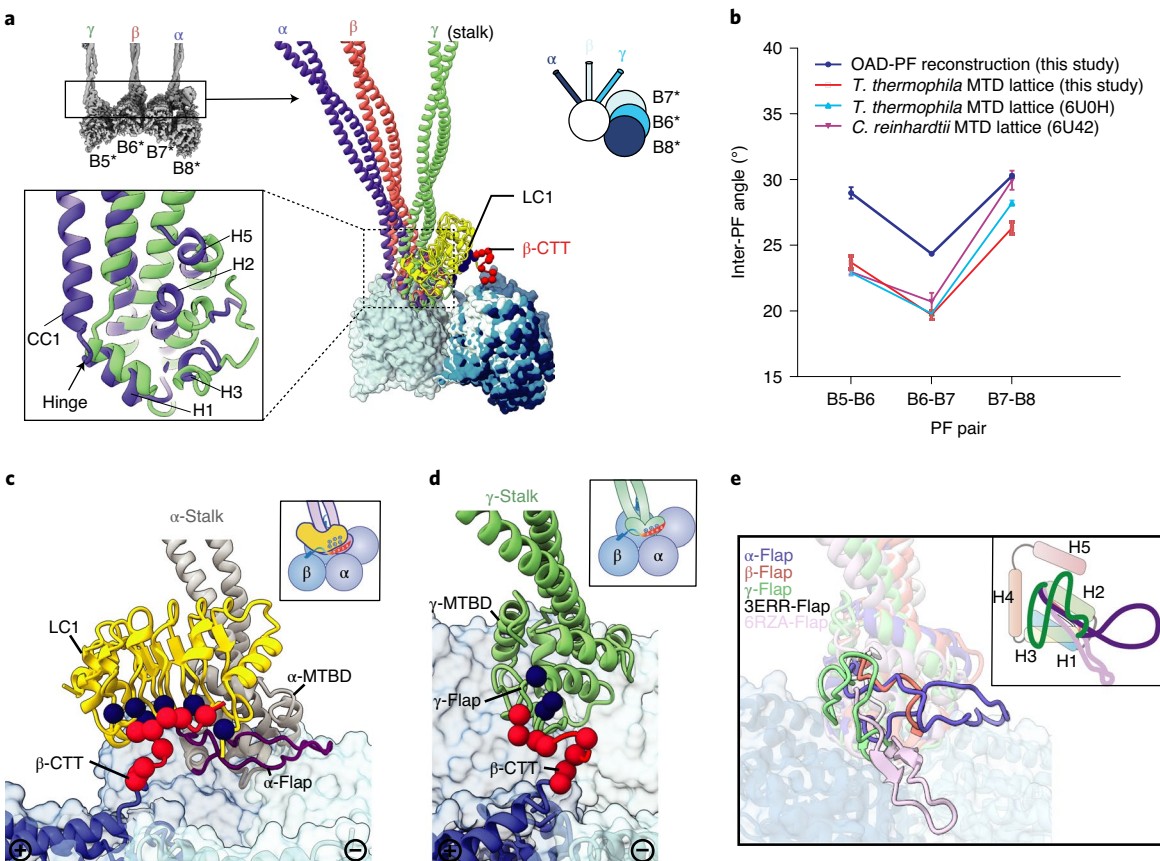

**Fig. 3 | Structures of the three distinctive MTBDs and their interactions with MTD. a**, A structural comparison among α-, β- and γ-MTBD, together with the stalks. Structures are aligned by superimposition of the three MTBD-bound protofilaments. A detailed comparison between α- and γ-MTBD is shown in the lower left inset. The relative rotations between adjacent protofilaments are illustrated with a model at the upper right. The white circle indicates the three aligned MTBD-bound protofilaments (B5–B7). The light blue, blue and dark blue sticks indicate the β-, γ- and α-stalk, respectively. The four protofilaments in the reconstituted OAD-PF may not necessarily be the native B5–B8 (*). **b**, The inter-PF angle pattern in the OAD-PF reconstruction is similar to that in native B5–B8 PFs from three independent MTD studies. Each inter-PF angle was estimated from three different axial positions. Data are presented as mean values ± s.d. **c**, The negatively charged CTT (red spheres) of the adjacent β-tubulin is attracted to the positively charged (blue spheres) surface of LC1. **d**, The interaction between the γ-flap and β-CTT. **e**, A comparison among the five flaps, including the three from our OAD structure, a low-affinity cytoplasmic dynein-1 MTBD (PDB 3ERR, white)[39] and the recently published MTBD structure of DNAH7 (PDB 6RZA, pink)[38]. A cartoon model is attached at the upper right.

inter-PF angles from our OAD-PF reconstruction with that of the native MTD protofilaments B5–B8. The overall patterns strikingly match each other (a drop of inter-PF angle in B6–B7 and an increase in B7–B8) (Fig. 3b)[15,36], implying a preferred inter-PF pattern for binding the three MTBDs together. This is likely to be a result of the distinctive structures of the three MTBDs. The α-MTBD specifically binds the conserved LC1 with its helix H5 (Fig. 3a,c), in line with a recent crystal structure[34]. The α-MTBD/LC1 complex requires a wider inter-PF space (similar to that between B7 and B8) to properly interact with adjacent tubulins (Fig. 3b). This interaction is mediated by a cluster of positively charged residues[37] on the LC1 surface that attract the negatively charged C-terminal tail of β-tubulin (β-CTT) from an adjacent PF (Fig. 3c and Extended Data Fig. 6d). Meanwhile, LC1 binding forces the α-flap (a variable loop between MTBD H2 and H3) to fit a conformation without contacting β-CTT (Fig. 3c). By contrast, the γ-flap directly contacts β-CTT with a similar charge–charge interaction (Fig. 3d and Extended Data Fig. 6e). The additional charge–charge interactions between LC1 and β-CTT may enhance the microtubule-binding affinity of OAD. This contrasts with the interaction between the flap of inner-arm dynein DNAH7 and its adjacent protofilament, which generates distortion of the microtubule singlet[38] (Fig. 3e). The β-MTBD has

a much shorter flap (Fig. 3e) and lacks density connection with β-CTT, similar to that in dynein-1 (ref. [39]). These distinct features of the three MTBDs collectively facilitate the landing of OADs onto MTD with a local inter-PF angle pattern similar to that of native binding sites.

**OAD array formation via the TTH interaction.** We then sought to elucidate the structural basis for how the OAD array is formed. In the parallel OAD, the two ICs directly contact and cross over each other at multiple sites (Extended Data Fig. 5c), which facilitates core-OAD to locally form a tight heterotetrameric architecture around the WD domains (Fig. 4a). The two ICs and N-terminal dimerization domain (NDD) of α/β-HCs (IC-NDD region) together enclose and position the tail region of one OAD (OAD$_0$), which is primed to interact with the motor domains of another OAD to the microtubule plus end (OAD$_{+1}$) (Fig. 4a). More OADs are subsequently induced to associate with each other in a TTH manner (Figs. 1d and 4a). We re-analyzed previously reported cryo-ET maps of axonemes and found all apo-OAD arrays are assembled in the same manner (Extended Data Fig. 7a)[9,10,12,13,40]. The TTH interfaces involve IC3, NDD and α/β-HB1–3 of the OAD$_0$ tail region, and α/β-linkers, AAA2 small subunit of β-HC (β-AAA2S),

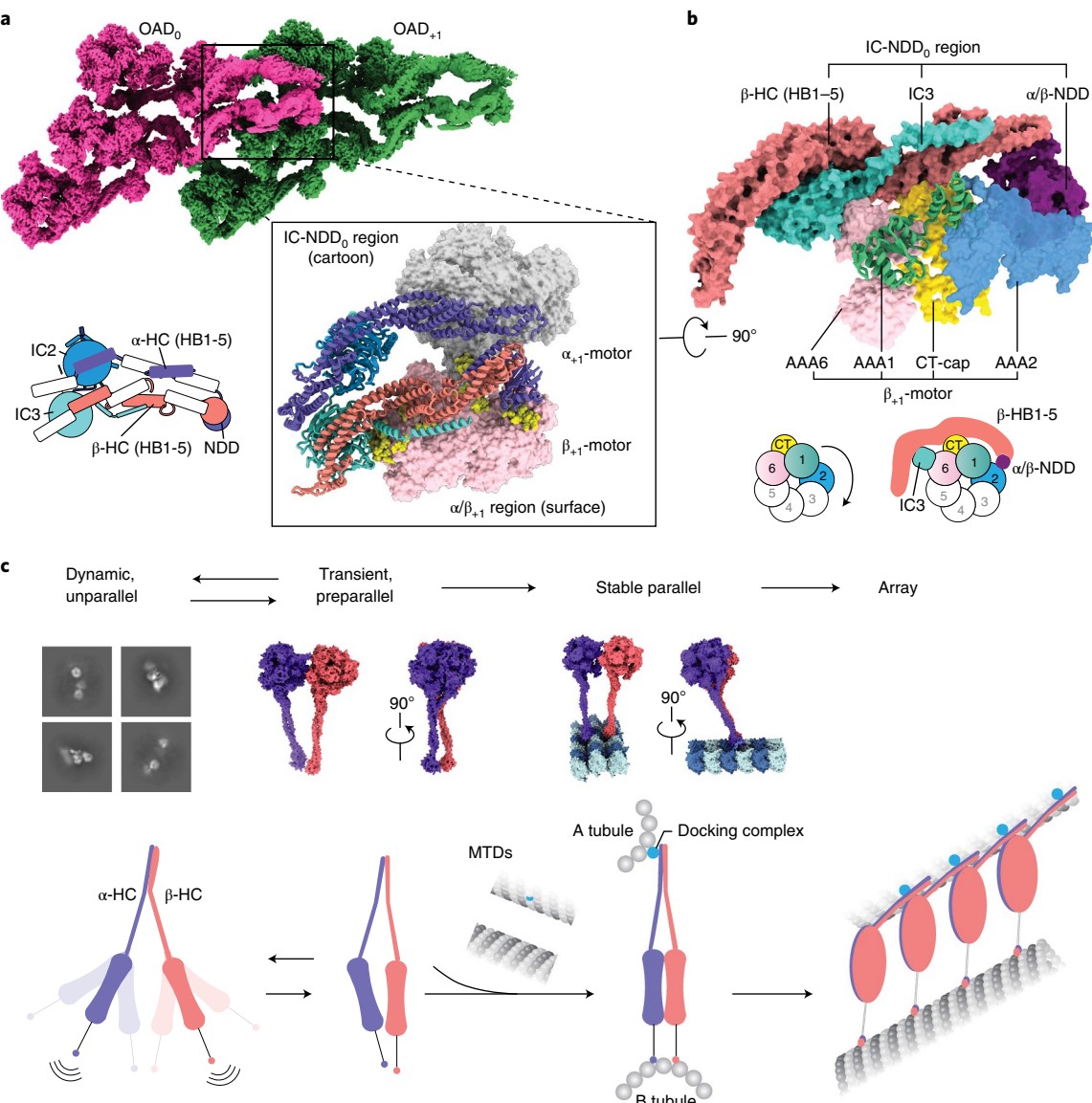

**Fig. 4 | OAD array formation via the TTH interaction. a**, The IC-HC forms a compact claw-like conformation in the tail region and grasps the head region (mainly β-ring) of an adjacent OAD unit. The top view of the OAD array clearly shows the TTH interaction. Details are enlarged in the lower right panel. A schematic representation of the tail region of $OAD_0$ is shown at the lower left. **b**, A close-up view of the TTH region. The schematic shows how the $tail_0$ region calipers the $β_{+1}$ ring and restrains its free allosteric response. **c**, A model for the OAD array formation mechanism. The γ-HC and LCs are not included in the model as they do not contribute to the critical interactions between adjacent OADs.

β-AAA6S and the C-terminal cap of β-HC (β-CT-cap) of the $OAD_{+1}$ head region (Fig. 4b and Extended Data Fig. 7b,c). Notably, the $tail_0$ calipers the $β_{+1}$-motor and potentially hinders its allosteric response to nucleotides (Fig. 4b).

To further understand why microtubule binding is critical for the TTH interactions, we performed cryo-EM on free OADs. In the absence of MTD, most free OADs are too flexible to support a stable interaction between OADs (Extended Data Fig. 8a–d). However, there exists a notable preparallel conformation (Extended Data Fig. 8b), in which the motor domains are stacked next to each other. We obtained a free OAD structure in that state at resolutions of 5–12 Å for different regions (Extended Data Fig. 5e,f). This transiently adopted preparallel conformation is different from the fully parallel structure of arrayed OAD. It is characterized by the relative rotations among the three motors, along with the OAD stalks converging toward the MTBD region (Extended Data Fig. 5e). This conformation is likely to facilitate a proper landing of OADs onto

microtubules, but is not yet ready for binding another OAD. It is the MTD binding that finally aligns the three motor domains, induces the linker–ring interaction to stabilize a fully parallel architecture (Fig. 1e and Extended Data Fig. 4d) and triggers multiple OADs to cooperatively associate with each other (Figs. 1d and 4c). It is worth noting that although OAD arrays can spontaneously form in the presence of MTD in vitro, docking complexes are essential and play a critical role in anchoring OADs to the right location in the axoneme[41], which reinforces MTD protofilament recognition by the MTBDs.

**TTH interaction preservation by coordinated MTBS alteration.** The OAD array is dramatically remodeled when β-HC moves one step ahead from MTBS-1 to MTBS-2 (Supplementary Video 3), coupled with five major structural changes throughout the entire OAD-MTD complex. First, an 18-degree rotation between the β-stalk and β-MTBD is required for microtubule-binding

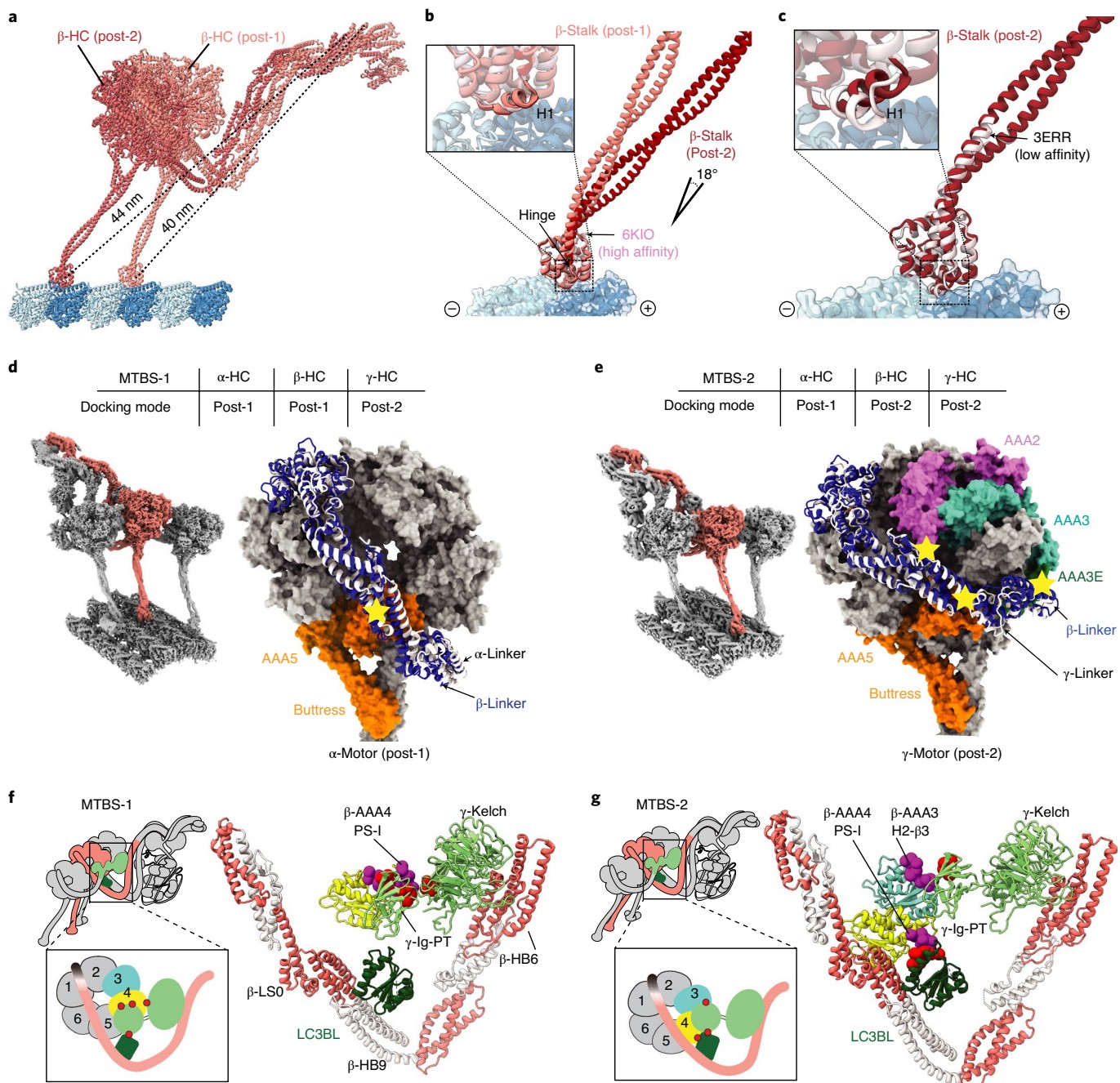

**Fig. 5 | Coordinated conformational changes of the OAD array from MTBS-1 to MTBS-2 for TTH preservation. a**, A comparison of β-HC structures in MTBS-1 and MTBS-2 by superimposition of the MTD protofilaments. **b**, The β-MTBD in MTBS-2 adopts the same conformation as that in MTBS-1 (upper left), while its stalk orients differently (18°). **c**, The β-HC in MTBS-2 combines the low-affinity stalk conformation (black arrow) and the high-affinity state of MTBD, as indicated by the helix H1 (enlarged square at upper left). **d,e**, The gear-shift-like switch of linker–ring interactions. β-Linker (blue) adopts post-1 (classical, same as α-HC) and post-2 (novel, same as γ-HC) docking modes in MTBS-1 (**d**) and MTBS-2 (**e**), respectively. **f,g**, Remodeling of the local interaction network around LC3BL. The AAA4 PS-I region switches its interaction with the γ-Ig-PT/kelch region in MTBS-1 (**f**) to LC3BL in MTBS-2, along with formation of a new contact between AAA3 and Ig-PT (**g**).

state alteration (Fig. 5a,b), whereas β-MTBD remains in the high microtubule-affinity state[42] (Fig. 5a,c). Second, a gear-shift-like switch of the linker–ring interactions among the three motor domains is involved. In MTBS-1, β-HC adopts a post-1 linker docking mode (Fig. 5d), which is precisely switched to post-2 along with β-MTBD stepping forward (Fig. 5e). On the other hand, α-linker fits well into a groove (groove-1) formed by AAA3S, AAA5E and β-CT-cap in MTBS-1 (Extended Data Fig. 9a), while

β-linker matches another groove (groove-2) between γ-AAA2S and γ-AAA3S (Extended Data Fig. 9b). From MTBS-1 to MTBS-2, the α-linker docking groove on β-ring is also switched from groove-1 to groove-2 (Extended Data Fig. 9c). Third, the local network among β-tail, γ-tail and LC3BL is remodeled. In brief, γ-Ig-PT contacts β-AAA4L PS-I with the help of γ-kelch in MTBS-1 (Fig. 5f), while the contact site is switched to the H2-β3 loop of β-AAA3L in MTBS-2 (Fig. 5g and Supplementary Video 3). From MTBS-1 to

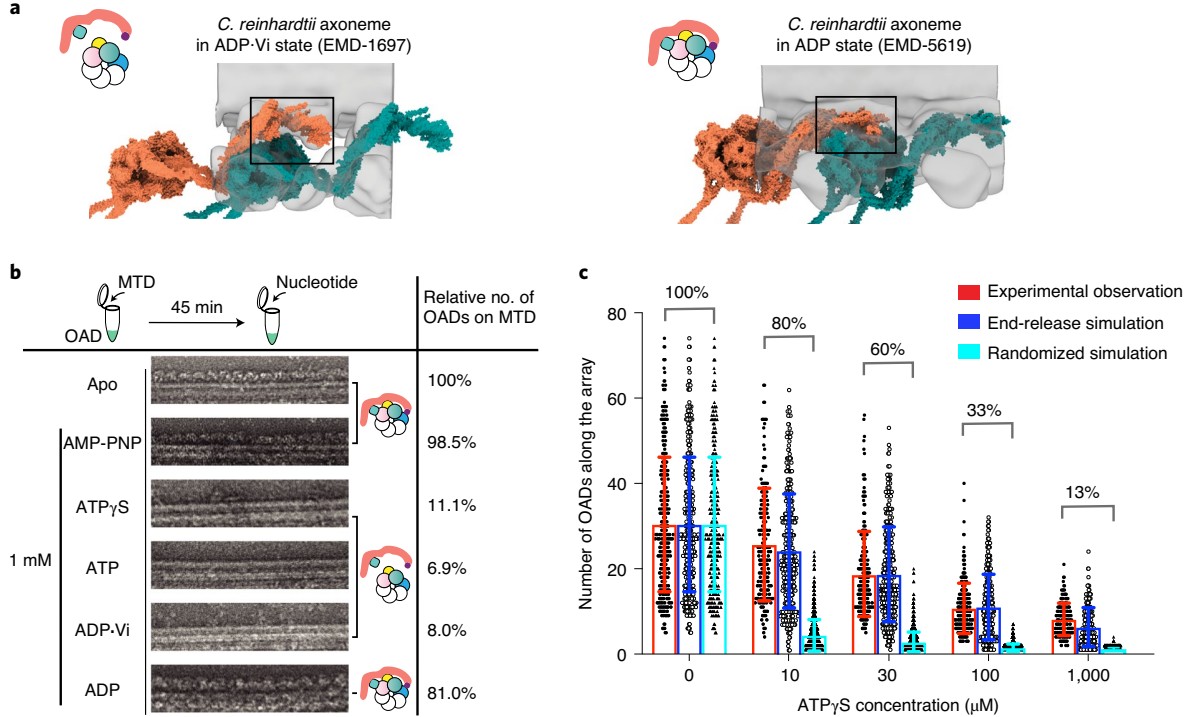

**Fig. 6 | Changes of TTH interfaces deliver the allosteric signal for motor coordination. a**, Different conformations of OAD array during the ATP cycle. The models (tomato and teal surfaces) were built based on previously reported cryo-ET maps[11,17] (transparent gray) and our OAD coordinates. The TTH interfaces in different nucleotide states are highlighted in the black squares, each illustrated with a cartoon model. **b**, The effects of different nucleotides on the reconstituted OAD-MTD arrays in vitro. ATPγS, ATP and ADP·Vi disrupt almost all arrays, but AMP-PNP and ADP do not. The models illustrate how different nucleotides affect the TTH interfaces. Total OAD unit number of all arrays in the apo state was used as a reference and normalized as 100% for subsequent comparison. The percentages represent the relative OAD coverage (including individually bound OADs) on MTDs in different nucleotide conditions compared to the apo state. **c**, The effects of ATPγS on the length of OAD array at different concentrations. OAD number on each continuous array from negative-stain images was manually counted and showed as a single black dot (left) at ATPγS concentrations of 0 μM (n = 319 microtubules), 10 μM (n = 180), 30 μM (n = 214), 100 μM (n = 222), 1,000 μM (n = 159). Error bar represents s.d. Gray circles (middle) and black triangles (right) in each column represent computationally simulated distribution of the array lengths by end-release (middle) or stochastic disruption (right) of OADs from the arrays to the same coverage ratio as that of the experimental observation. The start of both simulations is identical to that of the experimental data (0 μM ATPγS). Source data for **c** are available online.

MTBS-2, LC3BL takes over γ-Ig-PT to contact β-AAA4L. Fourth, the tail of OAD is rotated downward with respect to the head from MTBS-1 to MTBS-2 (Supplementary Video 3). Finally, in contrast to the nearly straight PFs in MTBS-1, they are coordinately bent by the OAD array in MTBS-2 (Extended Data Fig. 9d).

Comparing with previously reported dynein structures reveals that the ADP-bound state[43] has a similar linker docking mode to post-2 of β-HC. The ADP-bound state is thought to represent the rebinding of dynein MTBD to the microtubule after one step[44]. Therefore, the β-HC in post-2 probably mimics the state after one complete nucleotide cycle. After ATP treatment on the native *T. thermophila* axonemes and reverting to ATP-free solution, we could observe both post-1 and post-2 states of OAD arrays on the same cryo-ET data (Extended Data Fig. 9e,f). Despite the remodeling of the OAD array from MTBS-1 to MTBS-2, the TTH interactions remain nearly unchanged (Extended Data Fig. 9g,h). In either MTBS, the core-OAD conformations are synchronized along the same array (Extended Data Fig. 9e,f). However, the conformations in the two states are not compatible. Steric clashes are unavoidable by swapping their OAD units (Extended Data Fig. 9i,j). Therefore, the TTH interfaces need to be temporarily disrupted to complete the MTBS alteration. Most importantly, this is unlikely to occur within a locally synchronized OAD array except at the ends or a transition point where the TTH is temporarily relaxed.

**Changes of TTH interfaces are required for OAD coordination.** We then ask what are the key factors that potentially affect the TTH interfaces. We propose two possibilities, nucleotide control and geometry control. First, we re-analyzed previously reported cryo-ET structures of core-OADs by fitting our atomic coordinates into the density maps. Interestingly, TTH interactions in the pre-powerstroke state are quite different compared to the post-powerstroke states, as the β+1-motor is thoroughly released from the IC-NDD0 region along with a clockwise and downward rotation. This conclusion is consistent across species (Fig. 6a and Extended Data Fig. 10)[12,17,40]. We then tested how the reconstituted OAD-MTD arrays respond to different nucleotides. Briefly, the OAD arrays could still be observed after incubation with either AMP-PNP or ADP, but fell off the MTDs in the presence of ATP, ADP·Vi or ATPγS (Fig. 6b). Therefore, it is the ATP hydrolysis that releases the MTBDs from MTD and subsequently breaks the TTH interfaces along the array. Using a titration of ATPγS, we were able to gradually shorten OAD arrays in vitro owing to the low hydrolysis rate (Fig. 6c). Increasing the ATPγS does not cut the array into many shorter segments randomly, but rather sequentially shortens the arrays, suggesting ATP hydrolysis is more likely to take place at the ends of an ordered array and is inhibited for the rest of the OAD. We simulated the process under two different assumptions, random break and sequential fall-off. The simulation results indicate that it is more likely to be a sequential process. This is because the tail

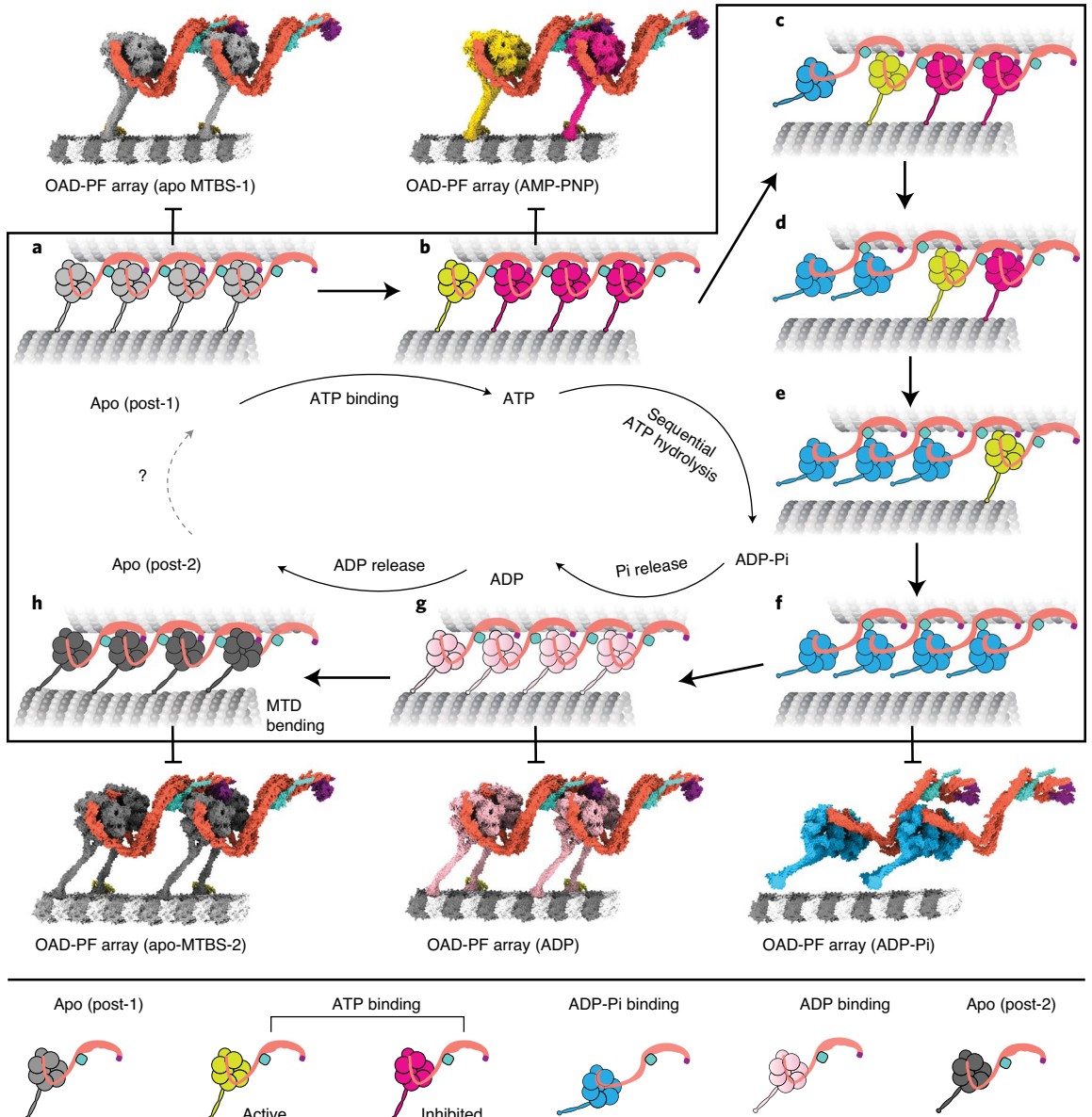

**Fig. 7 | A proposed model for the allosteric switch role of TTH interfaces in motor coordination. a**, An OAD-MTD array adopts post-1 in the apo state. **b**, In the ATP-bound state, the OAD unit that has relaxed motor domains is active (yellow), while the rest (magenta) are restrained by TTH interactions. **c–f**, Once the TTH interfaces are disrupted by ATP hydrolysis of the active OAD, it will sequentially relax downstream OAD units (second unit relaxed (**c**), third unit relaxed (**d**), fourth unit relaxed (**e**), all units in the pre-powerstroke state (**f**)) along the array. **g**, A putative model of the OAD-MTD array in ADP-bound state, which is presumed to be 8 nm forward. **h**, After ADP release, the OAD array adopts the post-2 state and bends the local MTD. The models of the OAD array in ADP-Pi and ADP states were built by fitting our atomic coordinates into previously reported cryo-ET maps (EMD-5758, EMD-5619)[12,17].

region from $OAD_0$ hinders the ATP-dependent allosteric response of $OAD_{+1}$ (Fig. 4a,b). Therefore, the changes of TTH interfaces between two adjacent OADs play a key role in delivering allosteric signals to allow motor coordination.

## Discussion

Cilia-driven movement of cells is among the most fundamental cellular activities during evolution. The rhythm of ciliary beat is defined by coordinated actions among OADs[19]. In this study, we focus on exploring the motor coordination between adjacent OADs of the array. On the basis of our high-resolution cryo-EM structures and previous cryo-ET studies[9,12,13,17,40], we propose the following model to explain how arrayed OADs coordinate with each other to take one step (Fig. 7).

In post-1, OADs are attached to B-tubules and form ordered arrays (Fig. 7a). The TTH interactions between $OAD_0$ and $OAD_{+1}$ sterically restrain the nucleotide-induced allosteric response of $OAD_{+1}$ and its downstream neighbors (Fig. 7b). Various mechanochemical factors may contribute to the initiation of local TTH interface relaxation. It is possible that local curvature changes may be involved in the regulation of dynein-MTD affinity. After initial relaxation of the TTH interfaces, $OAD_0$ is free for its nucleotide cycle and the energy from ATP hydrolysis is consumed to release $OAD_0$ from B-tubule, rotate the motor domain and move it one step forward (Fig. 7c). Due to the forward movement, this in turn relaxes the TTH interfaces between currently active $OAD_{+1}$ and its downstream neighbors (Fig. 7d–f). The process is rapidly propagated to the plus end of MTD, consistent with the observed

powerstroke rate of dynein[45] and the fact that most OADs adopt the pre-powerstroke state in cilia in the presence of ATP[18]. Meanwhile, the release of phosphate groups and ADPs from upstream OADs will lead to rebinding of those motors to B-tubules and re-formation of TTH interactions between adjacent OADs in post-2 (Fig. 7g). Since the distances between the tail and MTBD regions are changed from post-1 to post-2, and the tail region is permanently anchored to MTD A-tubule, preservation of the TTH interactions between the two states will subsequently impose tension between two adjacent MTDs. The tension is converted into the local bending of MTD (Fig. 7h), which is propagated toward the plus end along with the nucleotide cycle propagation. Our current cryo-EM structures and the proposed model potentially explain how a rhythmic ciliary beat is generated through phased propagation of OAD nucleotide states and microtubule-binding states.

It was previously proposed that dynein rebinds to microtubule in a hypothesized pre-powerstroke II (pre-II) state and reverts to the post-powerstroke state (post-1) along with an 8-nm movement of the whole motor after release of ADP and phosphate groups to generate a sliding[12]. Our cryo-EM structures strongly suggest it is more likely to be post-2 that represents the MTD-rebound state immediately after one complete nucleotide cycle. It is possible that the post-2 state observed in our study occurs after the previously proposed pre-II state[12]. However, our assays on the effects of nucleotides suggest this is unlikely due to the low affinity between MTD and OAD with ATP bound. The final conformational changes of the motor domain from post-1 to post-2 are produced by a relative rotation between the AAA+ ring and linker (Fig. 5d,e), while the OAD motors remain in the same position without moving forward (Fig. 5a and Extended Data Fig. 9e–h). Cytoplasmic dyneins[44,46] and OADs seem to share a universal mechanism for tension generation to this step. However, in cytoplasmic dyneins, the tension is eventually consumed to move the cargo forward[46]. By contrast, OADs utilize the tension for MTD sliding (Extended Data Fig. 9d) as the tails are permanently anchored to the A-tubule. The anchored tail region in turn helps the motor domain maintain its original position after rebinding to MTD. This is critical for the preservation of the TTH interactions between the two states (Extended Data Fig. 9g,h) because large movement of the motor domains will otherwise lead to a conformation mismatch in the TTH interfaces. On the other hand, free nucleotide cycle of a downstream OAD requires temporal disruption of TTH interfaces, which can be achieved by ATP hydrolysis. Therefore, alternation between the disruption and re-formation of TTH interactions is a key process to ensure the motor coordination and MTD bending during the mechanochemical cycle of an OAD array.

Our work also provides key structural information of several light chains, which are important for sensing different signals[32,33,37,47–50]. For example, the LC4A equivalent in *C. reinhardtii* regulates the microtubule binding of OAD in an ATP-sensitive manner[32] and alters the conformation of γ-tail (equivalent to α-tail in *T. thermophila*). We speculate that the regulation is achieved through Ca²⁺-induced conformational change of LC4A. Our current structure represents the typical Ca²⁺-free state of calmodulin, in which the two EF-hand pairs are joined by the loose central linker (Extended Data Fig. 5f). Upon an influx of Ca²⁺, this joint linker of LC4A will be induced to form a helix[51], which is likely to impose tension between the α-tail and IC/LC-tower and affects the allosteric response of α-HC. Our structure also reveals that LC1 contributes to the MTD protofilament recognition via its interaction with adjacent β-CTT (Fig. 3a,c). It was previously suggested that β-CTT directly interacts with dynein and antibody binding decreases the flagellar beat frequency in sea urchin spermatozoa[52]. Mutations of the positively charged residues on LC1 surface also lead to a disruption of microtubule binding in *C. reinhardtii*[37]. On the other hand, post-translational modification of the β-CTT is critically important

for cilium assembly and beating regulation[53]. All these imply that interaction between β-CTT and LC1 is important for regulating OAD activity. During ciliary beating, the orientations of the charged surface of LC1 vary along with MTD bending, which will probably affect interactions between LC1 and β-CTT. Therefore, our structural observation supports the previous hypothesis that the affinity between MTD and α-MTBD is regulated in a curvature-dependent manner[34,54]. Considering LC1 and β-CTT are conserved across species, beating regulation through their interaction is likely to be a universal mechanism. The thioredoxin LC3BL is localized in an intricate local network formed among β-motor, β-neck and γ-tail (Extended Data Fig. 5b). We show that LC3BL switches its contacts with β-HC and γ-HC during MTDB alteration (Fig. 5f,g) and potentially coordinates the conformational changes between the two heavy chains.

Our work demonstrates that it is now possible to understand such an intricate subcellular system of the OAD arrays on MTD in near-atomic detail. In future, it will be interesting to clarify how IADs, N-DRCs, RSs and CPC interplay and collectively regulate the OAD array to accomplish a rhythmic beat.

## Online content

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

## Methods

**Purification of OAD and MTD.** Mucocyst-deficient strain *T. thermophila* SB715 and wild type (CU428) were purchased from Tetrahymena Stock Center (Cornell University, https://tetrahymena.vet.cornell.edu/). The cell lines were cultured in SSP medium and maintained at 130 r.p.m. and 30 °C. The axoneme was purified by using a modified dibucaine method from 4 liters of culture for each sample preparation. In brief, the pellet from every one-liter fresh cell culture was deciliated with 3 mM dibucaine (Sigma-Aldrich) in 150 ml fresh SSP medium and centrifuged at 2,000g for 10 min to remove the cell body. The cilia were spun down from the supernatant at 12,000g for 10 min, resuspended by axoneme buffer (20 mM HEPES pH 7.4, 100 mM NaCl, 2 mM MgCl₂, 1 mM DTT) and further demembranated with 1.0% Triton X-100 in axoneme buffer. The axoneme was then pretreated with buffer containing high potassium acetate (HPA buffer: 50 mM HEPES pH 7.4, 600 mM CH₃COOK, 5 mM MgSO₄, 0.5 mM EGTA, 1 mM PMSF, 1 mM DTT) for 30 min. Subsequently, the purified axonemes were treated under different conditions for different assays. For OAD purification, the axoneme was treated with high salt buffer (HSC buffer 50 mM HEPES pH 7.4, 600 mM NaCl, 5 mM MgSO₄, 0.5 mM EGTA, 1 mM PMSF, 1 mM DTT) and incubated on ice for 30 min. MTD and the majority of axonemal dyneins were separated by centrifugation at 21,000g for 10 min for further purification.

To obtain high-quality OAD complexes from the supernatant we used an OAD extraction protocol and optimized the parameters in our own experiments. The supernatant was laid over 5–25% (w/v) linear sucrose gradients in OAD buffer (50 mM HEPES pH 7.4, 100 mM NaCl, 1 mM DTT) and centrifuged at 153,000g for 16 h at 4 °C. The gradient was fractionated into 0.2-ml aliquots. Fractions containing IADs and OAD were determined by SDS–PAGE (4–20% Mini-PROTEAN TGX precast protein gels run in SDS buffer (Bio-Rad)), stained by SYPRO Ruby or Page-Blue staining solution. The pooled fractions containing the OAD were dialyzed (Repligen dialysis membranes) against the OAD buffer for 6 h at 4 °C to remove the sucrose and then loaded on an EnrichQ 5/50 column (Bio-Rad) equilibrated with ion-exchange buffer A (50 mM HEPES pH 7.4, 50 mM KCl, 1 mM DTT). The OAD was eluted with a linear salt gradient: 0–50% buffer B (with 1 M KCl) in 8 ml. The fractions containing OAD were determined by SDS–PAGE (Bio-Rad), pooled together and adjusted to a final concentration of 0.3 mg ml⁻¹ for reconstitution assay.

The MTD pellet was resuspended in a high salt buffer and dialyzed against the low salt buffer (50 mM HEPES pH 7.4, 0.5 mM EDTA, 1 mM DTT) overnight at 4 °C (ref. [55]). The MTD was then pelleted and resuspended with the fresh low salt buffer. The final concentration was adjusted to ~0.6 mg ml⁻¹ for the subsequent reconstitution assay. Grafix method[56] was used to improve the quality of free OAD samples for EM analysis. Briefly, the OAD fractions from the linear sucrose gradient were collected, dialyzed against the OAD buffer, concentrated to 1 ml and applied to Grafix (0–0.0125% (v/v) linear glutaraldehyde (Sigma-Aldrich) gradients along with a 5–25% (w/v) linear sucrose gradient in the OAD buffer). The gradient was fractionated into 0.2-ml aliquots. The cross-linking was quenched by adding 10 μl Tris–HCl (1M, pH 8.0) to each aliquot. The fractions containing the cross-linked OAD were determined by SDS–PAGE (Bio-Rad) and evaluated by negative-stain EM. The fractions containing properly cross-linked OAD were dialyzed against the OAD buffer for 6 h, concentrated to 50 μl and loaded to the TSKgel G4000SW$_{XL}$ column (Tosoh Bioscience) equilibrated with the gel filtration buffer (20 mM HEPES pH 7.4, 100 mM KCl, 1 mM DTT, 2 mM MgSO₄). The peak fractions were pooled for subsequent negative-stain analysis. Protein concentrations were measured using a BioSpectrometer (Eppendorf).

**Mass spectrometry.** Mass spectrometry (MS) on the isolated OAD sample was performed at Keck Biotechnology Resource Laboratory, Yale University. The OAD subunits identified from the MS data are summarized in Supplementary Table 1.

**Microtubule-gliding assay and analysis.** Dynein-gliding assay was adapted from a previously published protocol[57]. In brief, HMDE buffer (30 mM HEPES–KOH, 5 mM MgSO₄, 1 mM EGTP, 1 mM DTT, pH 7.4) was first introduced into the flow channel, followed by a 5-min incubation of 10 μl, 0.1 mg ml⁻¹ purified outer-arm dynein at room temperature to allow adsorption of dynein to the cover glass surface. Unbound dynein was then washed with HMDE buffer, followed by a 5-min incubation of 0.4 mg ml⁻¹ casein. The channel was again washed by HMDE buffer. GMPCPP-stabilized microtubules were prepared as previously described[58] using bovine brain tubulin purified in-house. A 10-μl portion of microtubule solution (~0.15 μM tubulin dimer in HMDE + 1 mM ADP) was perfused in to bind to the motors, with a subsequent wash by HMDE + 1 mM ADP. A 10-μL portion of motility solution (HMDE + 1 mM ATP + 1 mM ADP) was then flowed in to initiate the microtubule gliding, imaged by interference reflection microscopy with a frame rate of 13.5 Hz. Lengths and positions along the gliding paths of individual microtubules were tracked with the tracking software FIESTA[59] after background subtraction. Tracking results were manually inspected to exclude immobile filaments, surface dirt particles, tracks less than 1 s and tracking errors due to filament collisions. The position of individual microtubule filaments was averaged over three frames (0.22-s interval) to reduce the experimental noise. Time-weighted average velocity and displacement-weighted average velocity were calculated with the bin width of 0.4 μm s⁻¹. The standard error of the mean (s.e.m.)

of the displacement-weighted average velocity is equal to the standard deviation (s.d.) divided by $\sqrt{N}$, where $N$ is the number of microtubules in each condition ($N = 51, 75, 65, 78$ MTs for 100, 50, 20, 10 μg ml⁻¹ of OAD with wild-type GMPCPP microtubule, and $N = 51$ for 100 μg ml⁻¹ OAD with subtilisin-treated GMPCPP microtubule). The *P* value was calculated using Welch's *t*-test.

**OAD-MTD array reconstitution and nucleotide treatment.** The reconstitution condition was optimized from our cryo-EM analysis on the basis of a previously published protocol[16]. To assemble the OAD-MTD complex, the freshly purified native OAD (not cross-linked) and MTD samples were mixed at a series of molar ratios (tubulin dimer/OAD from 10 to 100) in the reconstitution buffer (20 mM HEPES pH 7.4, 100 mM KCl, 5 mM MgSO₄, 1 mM DTT) and incubated on ice for 45 min. The reconstituted samples were analyzed by SDS–PAGE and negative-stain EM to screen out the optimal molar ratio. The concentration of the OAD-MTD complex was optimized by centrifugation and resuspension for subsequent experiments and checked by cryo-EM. All protein samples were verified by 4–20% Mini-PROTEAN TGX precast protein gel stained with Page-Blue solution. The OAD array for nucleotide treatment was formed using excessive amount of OADs at a final concentration of 45 nM. Different nucleotides at a final concentration of 1 mM, or a titration of ATPγS at different concentrations, were added to the reconstitution tubes. A 4-μl portion of solution in each reaction condition was immediately transferred to a glow-discharged continuous carbon grid (Electron Microscopy Sciences) for 15 seconds before blotting and staining. The grid was stained with 2% uranyl acetate and air-dried before loading to a Talos L120C microscope (ThermoFisher Scientific). The length of the array was directly measured in ImageJ and converted to the equivalent number of OADs for subsequent quantitative analysis.

**Simulation of OAD fall-off.** We used the experimental data from OAD-MTD arrays in the apo state as the starting point for simulating the process for both end-release and stochastic fall-off. In the stochastic model, we regard all OADs in an array as able to equally access nucleotides at the same hydrolysis rate. In the end-release model, we assume that OADs at the ends of longer arrays may have equal or higher rates to fall off the microtubules. This assumption makes sense because a few OADs may simultaneously fall off, which occurs more frequently on longer arrays. To include all possibilities, we introduce a coefficient $\alpha$ and apply the following transform to the length of the *i*th array: $L_i = 1 + \alpha(N_i - 1)$, where $N_i$ is the OAD number of an array (*i*) and $\alpha$ ranges from 0 to 1. The probability that an end OAD of the *i*th array will fall off next is estimated as $L_i / \sum(L_i)$. When $\alpha = 0$, $L_i$ is constantly 1, which means all arrays with different lengths have the same probability to release OADs. When $\alpha = 1$, the probability that the next released OAD will appear on the *i*th array is proportional to its array length. We estimated the $\alpha$ by minimizing the discrepancies between experimental data and simulated results. The best estimation of $\alpha$ is ~0.1, which is quite close to zero, suggesting that the fall-off rate is weakly affected by the array length, and a longer array has only a slightly higher rate. We thus used $\alpha = 0.1$ for the final simulation of the end-release model.

**Cryo-EM sample preparation and data collection.** The 4-μl free OAD or OAD-MTD samples were applied to each Quantifoil R2/2 or C-flat R1.2/1.3 gold grid (for free OAD, the grids were coated with a carbon layer), incubated in a Vitrobot Mark IV (ThermoFisher Scientific) for 4 seconds, blotted for 2 seconds at 4 °C and 100% humidity and then plunged into liquid ethane near the melting point. Three cryo-EM datasets of OAD-MTD arrays in the apo state were collected on a 300-keV Titan Krios microscope (ThermoFisher Scientific) equipped with a Bioquantum Energy Filter and a K2 Summit direct electron detector (Gatan) at the Yale CCMI Electron Microscopy Facility. Data collection was automated by SerialEM software[60] and all micrographs were recorded in a super-resolution mode. The first two datasets were collected using the following parameters: 0.822 Å per pixel, 50 μm C2 aperture, 32 frames, 53.3 e⁻/Å², 8 s exposure, −0.8 to −2.0 μm defocus range. On the basis of the results of these two datasets, the third data acquisition was optimized with a reasonable parameter set as follows: 1.333 Å per pixel, 50 μm C2 aperture, 40 frames, 53.3 e⁻/Å², 12 s exposure, −1.2 to −3.0 μm defocus range. Three nonoverlapping micrographs per hole were recorded in all three datasets. Detailed data collection parameters are summarized in Table 1. The motion correction, particle picking and CTF estimation were streamlined to evaluate the micrograph quality in real time during the data collection using a modified preprocessing script (https://www2.mrc-lmb.cam.ac.uk/research/locally-developed-software/zhang-software).

**Cryo-ET data collection and reconstruction.** Purified axonemes were treated with ATP at a final concentration of 1 mM for 5 min and reverted to nucleotide-free solution before freezing for cryo-ET. In total, 50 tomographic datasets were collected on the 300-kV Titan Krios equipped with a K2 detector. The software SerialEM[60] was used for automatic data collection under the bidirectional scheme at a 3° interval and tilt angles ranging from −51° to +51°. Each of the final tilt series contains 35 movies with a pixel size of 2.8 Å at an average defocus of ~5 μm and a total dose of ~70 e⁻/Å². Individual movies were aligned by MotionCor2 (ref. [61]). Motion-corrected images of each tilt series were aligned by using the

patch-alignment approach in the IMOD software[62]. Subvolume average was performed using PEET[14].

**Preprocessing of cryo-EM data.** Beam-induced drift was corrected using MotionCor2 (ref. [61]) for all images. CTF parameters for each motion-corrected micrograph were estimated using Gctf[63]. All particles were automatically picked using Gautomatch, extracted in RELION v.3.0 (ref. [64]) and imported to cryoSPARC v.2.12 (ref. [65]) for all subsequent processing, if not explicitly stated otherwise.

**The MTD structure determination.** To obtain and analyze the structure of MTD, we first manually picked a small dataset (100 micrographs) at 4-nm intervals from dataset 1 (Table 1). These particles were analyzed in cryoSPARC v.2.12 to generate 20 good MTD 2D averages and used by Gautomatch for template-based particle picking. This generated 444,603 raw particles from datasets 1 and 2, and 680,495 raw particles from dataset 3 using a 4-nm distance cut-off (if the distance between two particles is less than 4 nm, the one with the lower cross-correlation coefficient is removed). All the particles were extracted with a box size of 512 × 512 pixels. The micrographs from dataset 1 and dataset 2 were both scaled to a pixel size of 1.333 Å to match dataset 3 during the particle extraction. After 3–5 cycles of 2D classification to remove those particles that generated bad 2D averages, the high-quality images were selected and filtered by a 6-nm distance cut-off. This reduced the sampling of MTD to ~8 nm and yielded 358,116 good particles for subsequent three-dimensional (3D) analysis. A previous MTD map from *T. thermophila* (EMD-8532)[55] was low-passed to 100 Å as an initial model. The 8-nm repeats were successfully separated into two classes of 16-nm repeats with comparable particle numbers after 3D classification in cryoSPARC v.2.12 (ref. [65]). The two 16-nm repeating maps were essentially the same except that they were shifted 8 nm with respect to each other. A total of 196,740 good particles with 16-nm periodicity were selected for subsequent analysis. By restricting the refinement to each local region with 3 × 4 tubulins, we were able to improve the local tubulins at an average resolution of 3.1 Å. A de novo model of the tubulin dimer was built on the best region and then expanded to all regions for manual refinement in Coot[66] and automatic refinement by REFMAC5 (ref. [67]). The tubulins from the 16-nm MTD repeat were used to estimate the inter-PF distribution.

**The OAD-PF structure determination.** To eliminate the interference of microtubules in the OAD structure determination, we linearly weakened the microtubule signals to improve the alignment of OAD. In brief, the coordinates of all good particles we selected during the MTD reconstruction were split and backtracked to their original micrographs. We manually checked all micrographs one by one to make sure they were centered and evenly spaced in each MTD. If not, we then manually adjusted the uncentered micrographs, added the missing particles or deleted the undesirable ones. The MTD signal was weakened by removing the weighted average within a rectangle mask slightly wider than the MTD. The OAD particles from the MTD-weakened micrographs were picked by Gautomatch using the 20 best templates generated from a negative-stain dataset of free OAD. After 2D classification, we selected the 50 best 2D averages for another cycle of automatic particle picking. Due to the severe orientation preference, we used a very low cross-correlation cut-off (0.08) and also a very small distance cut-off (150 Å) for automatic picking by Gautomatch. The purpose was to include as many views as possible at the beginning, even if there were some false pickings. This generated 824,659 particles from dataset 1 and 2, and 2,022,385 particles from dataset 3. Cycles of 2D and 3D classification (for screening purposes) were performed on the 8× shrunk images to remove MTDs and low-quality particles. In total, 346,320 good particles were selected for subsequent 2D and 3D analysis.

All particles from the above processing were re-extracted with a box size of 510 × 510 pixels at a pixel size of 1.333 Å (datasets 1 and 2 were rescaled to this pixel size) and merged for subsequent processing. To further remove particles that were less consistent with the major classes, we performed iterative 2D and 3D classification. Briefly, all the particles were separated into four subsets to accelerate the processing. Each cycle of 2D classification was followed by two cycles of 3D classification. A further 58,096 particles were excluded by means of this 2D and 3D classification. All the subsets were merged again, which yielded 288,244 good particles for a final cycle of 3D classification. This generated nine good classes and one bad class. Six of the nine classes were categorized to microtubule-binding state 1 (MTBS-1), while the remaining three were in MTBS-2. At this stage, we had 191,776 particles in MTBS-1 and 76,936 particles in MTBS-2 for subsequent local refinement.

A multilevel masking scheme was applied to the local refinement. Briefly, we gradually decreased the size of the mask applied to a certain region to ensure a stable local refinement. We divided each of the OAD-PF classes into five major parts: (1) MTBD-tubulin region, (2) α-motor domain, (3) β-motor domain, (4) γ-motor domain, (5) tail region. The α-motor domain in MTBS-1 was straightforwardly improved from a resolution of 10.1 Å to 4.5 Å after one cycle of local refinement. At such a resolution we were able to build backbones, but ab-initio assignment of the side chains was very challenging. The map was further improved by optimizing the following aspects: (1) more cycles of local classification, (2) refinement of the particle centers, (3) manual optimization of the local mask, (4) 2D classification based on 3D alignment parameters,

(5) local CTF refinement, (6) nonuniform refinement[68]. By combining these approaches in an iterative way, the map of the α-motor domain was finally improved to an average resolution of 3.19 Å for final model building. We applied the same strategy to improve the β-motor domain, which finally generated a map at an overall resolution of 3.3 Å. Focusing on the AAA2–4 subdomains allowed us to slightly improve this region, which helped a little with the model building. The γ-motor domain, MTBD region and tail region are much more complicated than the other two. We were able to overcome the issue by more levels of local refinement. To ensure that the maps from two adjacent regions can be smoothly combined, we applied a third mask that fully covered the boundary between each pair of adjacent masks. Finally, we integrated all 31 locally refined parts into an entire unit of OAD-PF array in Chimera[69]. The structures of free OAD and OAD-PF in the AMP-PNP bound state were determined using the same approach.

**Identification of the light chains.** We built the atomic model of all the ten IC-binding light chains de novo in combination with our MS data. First, each of the ten LCs was manually built as a poly(Ala) model. All side chains were tentatively assigned to several groups: (1) large (Trp, Try, Arg, Phe, His), (2) middle (Leu, Gln, Asn, Ile, Met, Lys), (3) small (Pro, Val, Ser, Thr, Cys, Glu, Asp, Ala) and (5) Gly. Here, we categorized Glu and Asp into the group 'small' because the side chain densities of negatively charged residues are typically weak in cryo-EM reconstruction. We then performed two parallel approaches to identify all the light chains: (1) pattern recognition and (2) penalty function. The first approach is based on regular expression match using the 'gawk' command on a CentOS 7.5 Linux system. In the second approach, we tried to fit all predicted homologs into a certain position, for example, the LC8-2b position, and assigned the residues. All the residues that did not match the side chain density were manually counted. The counts were regarded as penalty scores for all LC homologs. We then compared the final scores and selected the best one for subsequent model building and refinement. A protein was regarded as 'identified' only if it met the following requirements: (1) it exists as a significant hit from the MS data; (2) its side chains simultaneously match the cryo-EM density map; (3) no other homologs have better results of (1) or (2).

We identified IC2, IC3, γ-kelch and all ten IC-binding light chains. The LC7-a/b is not the standard LC7A/B heterodimer, but a heterodimer comprising LC7B (LC7-b) and an unnamed LC7A homolog (TTHERM_00348650). The full-length protein is 159 residues long (XP_976918.2), while the truncated one is 103 residues long (XP_976918.1). We unambiguously assigned the residues from S58 to G152. The extra density that links γ-HB6 to LC7-b was tentatively assigned as the N terminus of LC7-b.

Despite the similar core structures, each of the six LC8-like proteins (LC8s) were clearly different from any other five by their characteristic side chain densities and loops, which allowed us to distinguish them unambiguously. The positions of 1a, 1b and 2a are taken by LC10, DLC82 and LC8E, respectively. The remaining three (2b, 3a, 3b) were simply predicted to be LC8 homologs without standard names in TGD (TTHERM_00023950 for LC8-2b, TTHERM_01079060 for LC8-3a and TTHERM_000442909 for LC8-3b) (Supplementary Table 1). Neither TCT1A nor TCT1B matched the key features of our cryo-EM maps. The Tctex-a position was identified as a hypothetical homolog (TTHERM_00392979), while the best hit for Tctex-b is LC2A[49].

**Model building and refinement.** We used different model-building approaches for different regions. Most of the regions were refined at better than 3.5-Å resolution, which allowed us to build them in Coot[66,70] with side chains assigned and refined ab initio. For the regions that were slightly worse, we were able to build backbone models with the residues assigned on the basis of the relative positions among the large residues (such as Try and Arg) of each domain. For the regions that show clear backbone density with low-quality side chain density, we coarsely assigned the residues using previously published homologous structures as references or predicted models from the Phyre2 web server[71]. For those regions that were solved at a resolution with helices clearly separated, we fitted the predicted models into the density as rigid bodies in Chimera[69]. If the predicted model contained more than one subdomain (for example, LC4A), we then refined the fitting of each subdomain as a rigid body in Coot[70]. All models at better than 4-Å resolution were automatically refined by REFMAC5 (ref. [67]) followed by manual check in Coot[70]. The process was repeated until all parameters were reasonably refined.

**Inter-PF rotation angle measurement.** The inter-PF angle is defined as the lateral rotation angle between a pair of adjacent microtubule protofilaments, as described in a previous publication[36]. To estimate the inter-PF angles of MTD, we fitted individual tubulin dimers built from the 16-nm MTD reconstruction into the 48-nm MTD map as rigid bodies. We calculated the inter-PF angle between each pair of tubulin dimers from adjacent protofilaments using the 'angle_between_domains' command from PyMOL (https://pymol.org/2/). The averaged value and standard deviation were estimated from the three measurements in three representative regions of the MTD lattice, two regions close to the edges and one region in the middle.

**Visualization.** The figures and movies were created using Chimera[69], ChimeraX[72] and PyMOL (https://pymol.org/2/). Other tools used in this research include FIJI and EMAN2.

**Reporting Summary.** Further information on research design is available in the Nature Research Reporting Summary linked to this article.

## Data availability
The coordinates are deposited in the Protein Data Bank with PDB accession codes 7K58 (OAD-MTD in MTBS-1), 7K5B (OAD-MTD in MTBS-2), 7KEK (free OAD in preparallel conformation), 7N32 (four PFs of OAD-MTD), 7MWG (16-nm MTD), respectively. The cryo-EM maps are deposited in the Electron Microscopy Data Bank with accession codes EMD-22677 (OAD-MTD in MTBS-1), EMD-22679 (OAD-MTD in MTBS-2), EMD-22840 (free OAD in preparallel conformation), EMD-24066 (16-nm MTD). Source data are provided with this paper.

## Code availability
All scripts involved in cryo-EM data processing are available upon request.

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

## Acknowledgements
We thank S. Wu, K. Zhou, M. Llaguno and K. Li for technical support on microscopy; J. Kanyo for mass spectrometry support; Y. Xiong, F. Sigworth, J. Liu and S. Baserga for their valuable research advice; S. M. King, A. Yildiz, A. P. Carter and Y. Xiong for valuable feedback on the manuscript; and P. Sung, W. Konigsberg, A. Garen, Y. Xiong, as well as many others, for their generous support during the set-up of the K.Z. laboratory. This work was supported by start-up funds from Yale University, National Institutes of Health (NIH) grants (1R35GM142959) awarded to K.Z., and Rudolf J. Anderson Fellowship awards to L.H., Q.R. and Y.W.

## Author contributions
Q.R. prepared all samples and performed biochemical characterization. Q.R., K.Z., P.C., Y.W., L.H., R.Y. and Y.Y. determined the apo-OAD-PF structures. P.C., K.Z. and Q.R. determined all other structures in this work. Q.R., Y.-W.K. and J.H. performed and analyzed the motility assays. Q.R., Y.W., P.C., L.H. and F.H. performed other assays and analyses. All were involved in analyzing the results. K.Z., Q.R. and Y.W. prepared the manuscript with help from J.H. and other co-authors.

## Competing interests
The authors declare no competing interests.

## Additional information
**Extended data** is available for this paper at https://doi.org/10.1038/s41594-021-00656-9.

**Correspondence and requests for materials** should be addressed to Kai Zhang.

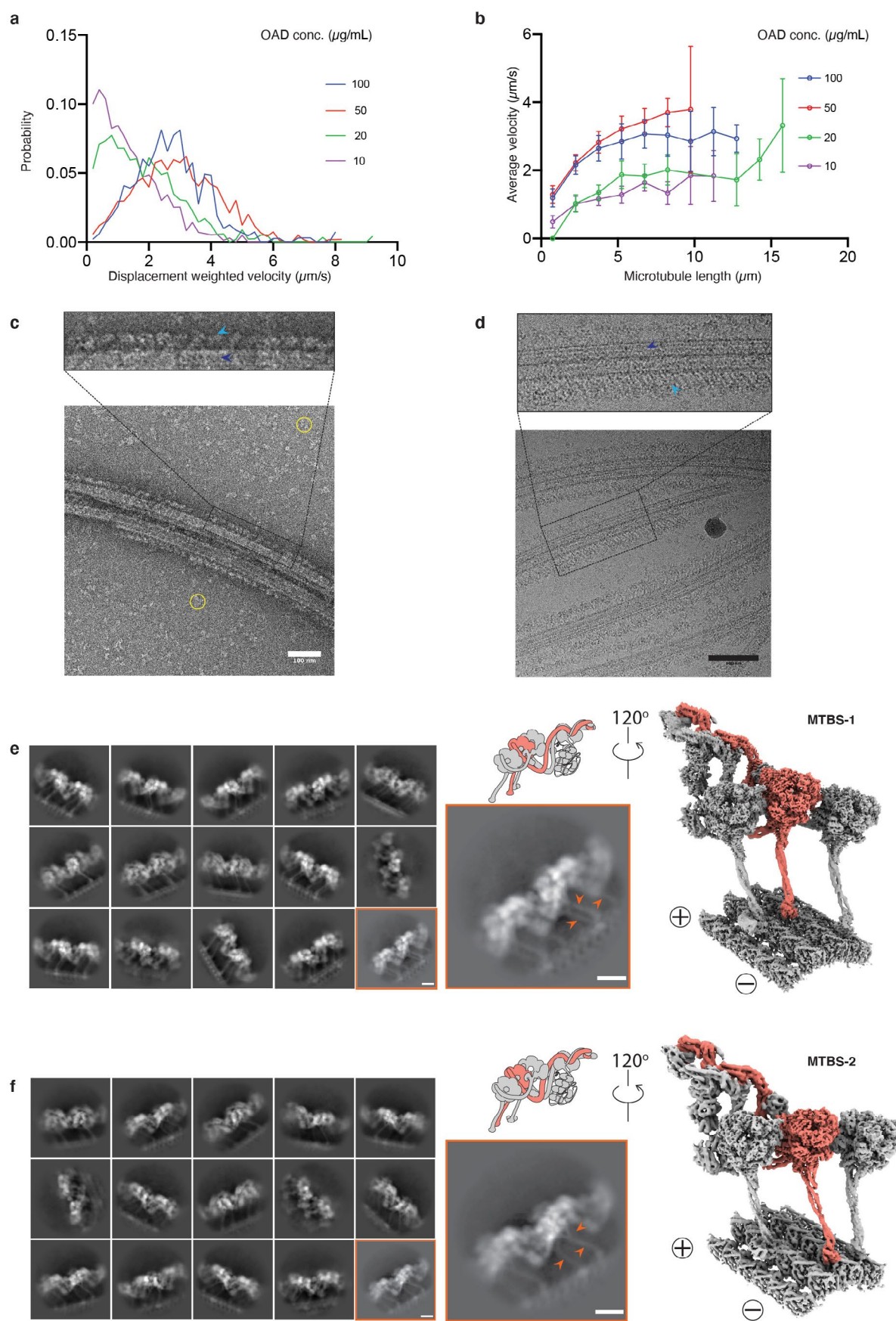

**Extended Data Fig. 1 | See next page for caption.**

**Extended Data Fig. 1 | Biochemical characterization and structure determination of OAD-MTD array. (a-b)** The gliding velocity of the free OAD on different lengths of microtubule. **(c-d)** The representative negative‾stain (**c**) and cryo-EM (**d**) micrograph of the reconstituted OAD-MTD sample show that the highly ordered OAD arrays with 24-nm periodicity are formed in the absence of nucleotides and docking complex, in line with previous findings[16]. The OAD arrays and MTDs are indicated by sky blue and blue arrows (upper), respectively. The excessive free OADs are marked by yellow circles in the background (before centrifugation). Similar images could be regularly acquired from n > 5 independent reconstitution assays. Scale bar, 100 nm. **(e-f)** The OAD arrays decorate MTD in two microtubule-binding states (MTBS-1 and MTBS-2). Representative 2D classes (left) and cryo-EM maps of the OAD-PF unit (right) in MTBS-1 (**e**) and MTBS-2 (**f**). The arrows indicate differences of the stalk orientations in MTBS-1 and MTBS-2 (**e, f**, lower middle).

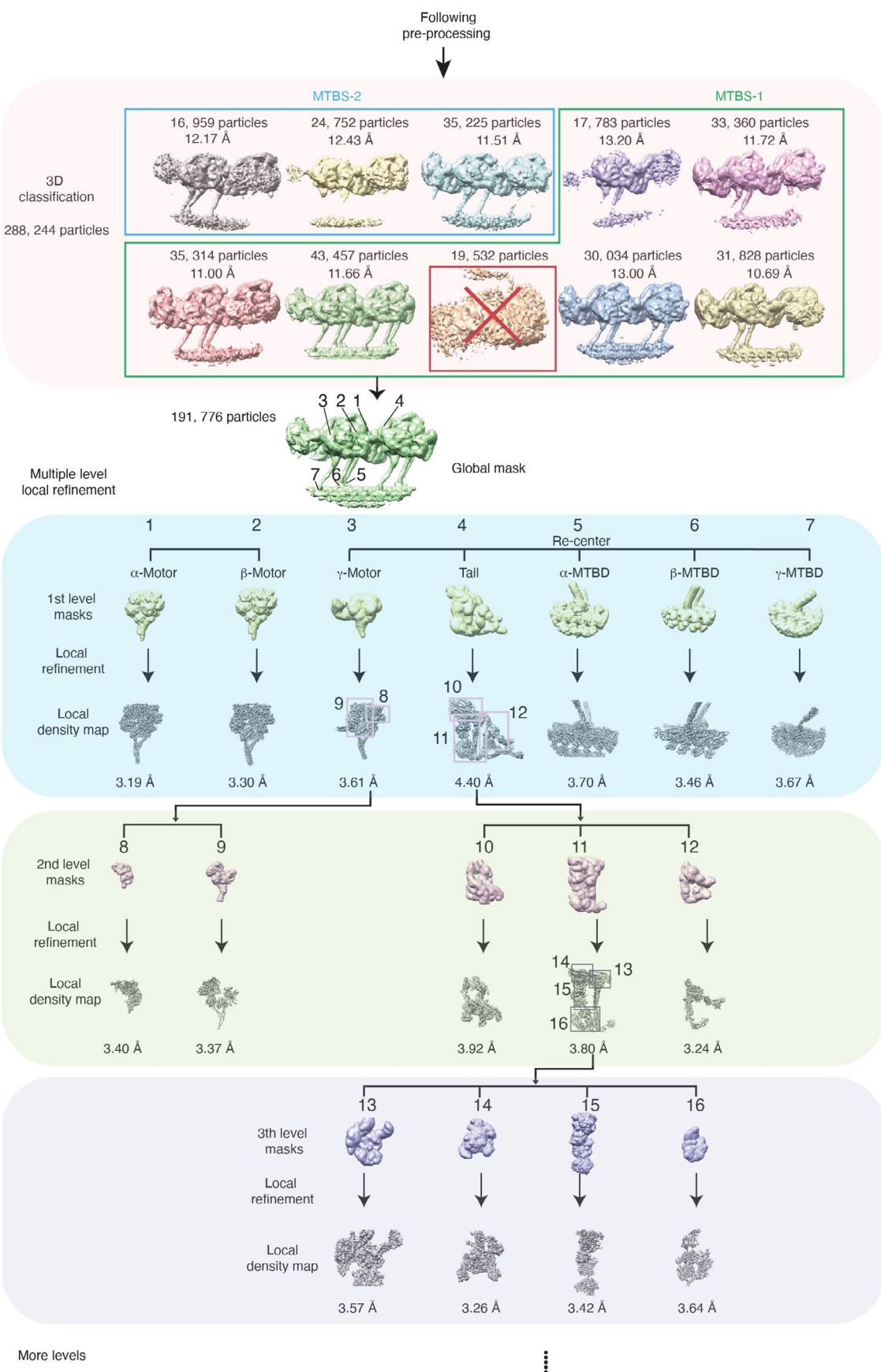

**Extended Data Fig. 2 | See next page for caption.**

**Extended Data Fig. 2 | Flow chart of the OAD structure determination.** The 3D classification identified two major microtubule-binding states (MTBS-1 and MTBS-2). The six good classes enclosed in the green rectangle are categorized into MTBS-1. The class in the red square was removed from the dataset for all following data processing. Three classes enclosed in sky blue rectangle are categorized into MTBS-2. Multiple levels of masks were implemented in the local refinement to improve the resolution of OAD-PF structure. The box size (~680 nm) was optimized to cover at least two adjacent OADs and the four PFs they bind to. The attached images represent typical local densities of the 3D volumes generated from cryoSPARC 2.12.

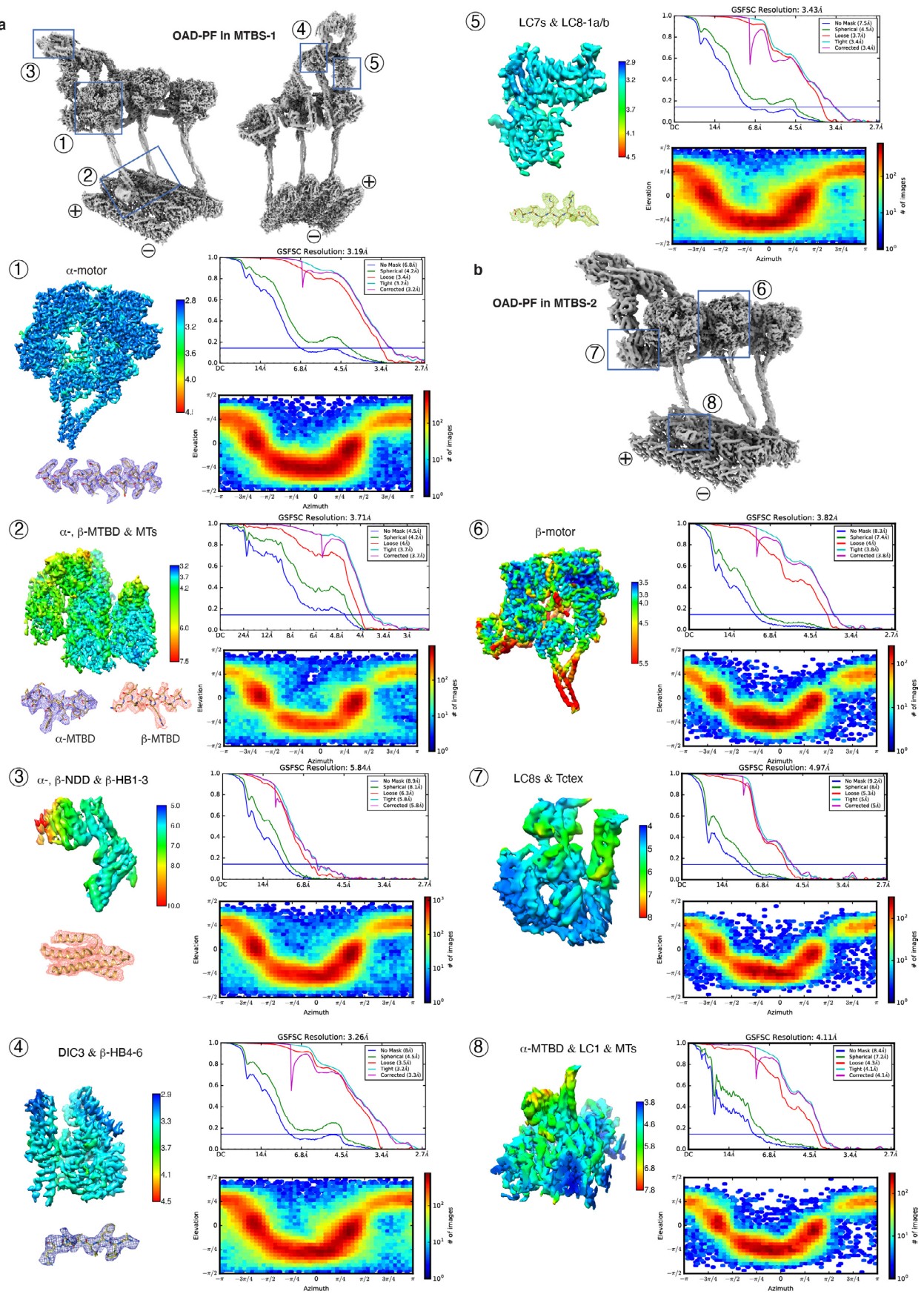

**Extended Data Fig. 3 |** Local resolution maps, FSC curves, orientational distribution and representative regions with the atomic model fitted into the local density for each mask from MTBS-1 **(a)** and MTBS-2 **(b)**.

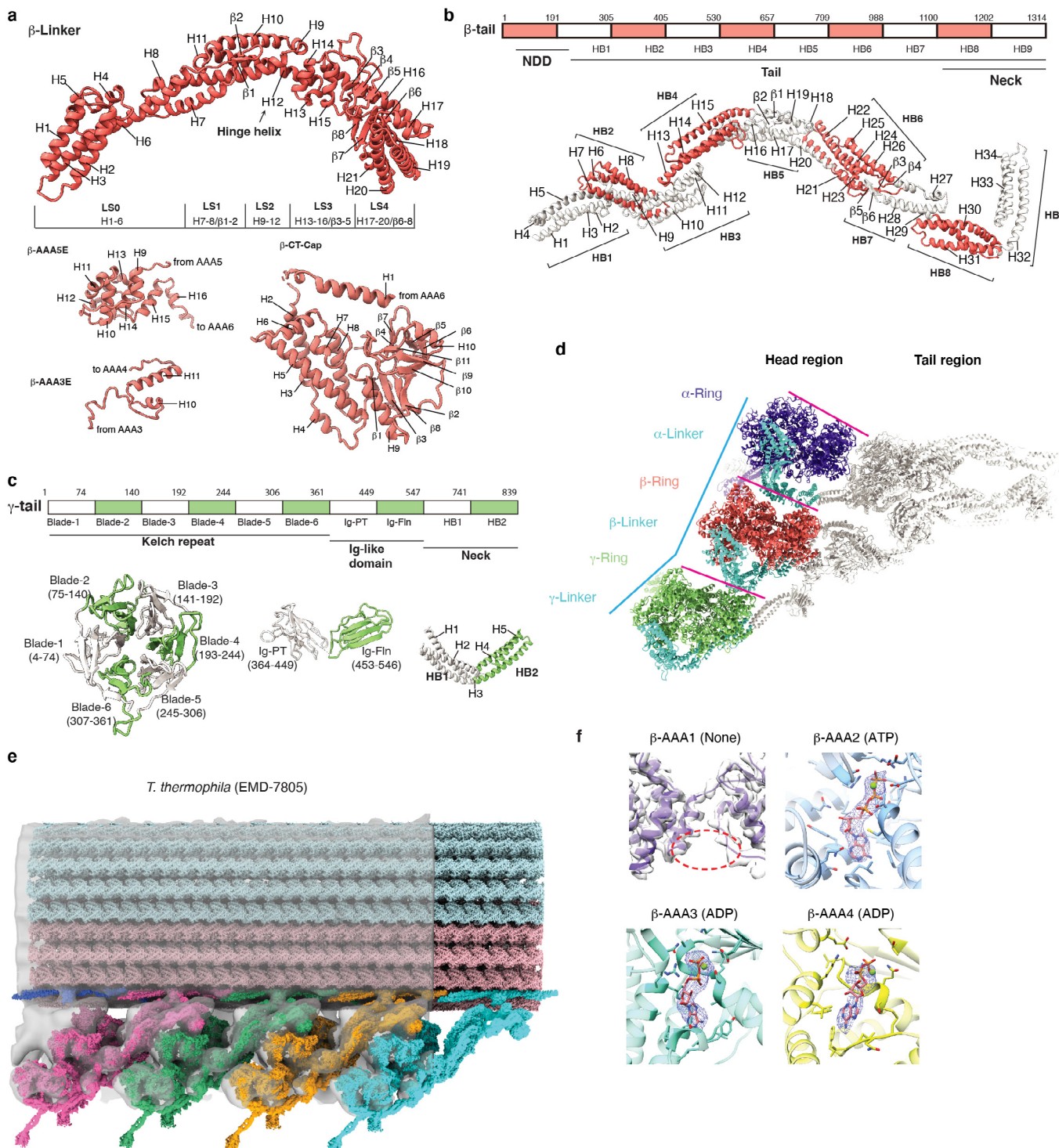

**Extended Data Fig. 4 | The parallel conformation of arrayed OAD in apo state. (a)** Detailed index models of the subdomains in the β-Linker region, motor domains AAA extensions (only the extensions used in this study are shown here), and CT-cap region. The subdomain of Linker and CT-cap of α-, γ-HC are the same as that of β-HC. The Linker helix H12 is the hinge helix[43] for the Linker and AAA + ring swing. LS, Linker subdomain. **(b-c)** Detailed index models of the subdomains of the tail region of β-HC (**b**) and γ-HC (**c**). The subdomain of α-tail is the same as that of β-tail. **(d)** Top view of the cryo-EM structure of OAD-PF unit shows that the three motor domains are organized in a nearly parallel architecture (the three pink lines). The relative positions of the α- and β-motor domains are aligned with the tubulin lattice. The center of the γ-motor domain is slightly ahead of the lattice line defined by connecting the centers of the other two (cyan polyline) in both MTBS-1 and MTBS-2. The OAD appears as a triangle shape from top view, with the β- and γ-tail twisting toward the α-tail with respect to the motor domains. **(e)** Rigid body fitting of our OAD array into cryo-ET maps of axonemes from *T. thermophila* (EMD-7805)[9]. The four OAD units were fitted into the cryo-ET density *en bloc* rather than individually which verified the TTH interaction. The N-terminal regions of the tails are also close to the docking complex and they are bridged by clear density connections. **(f)** Local densities of the nucleotides in AAA1-AAA4 of β-motor domain. Nucleotide density does not appear in β-AAA1. ATP in β-AAA2 and ADP in β-AAA3/AAA4 were unambiguously identified from the density. Other motor domains have the same nucleotide binding states.

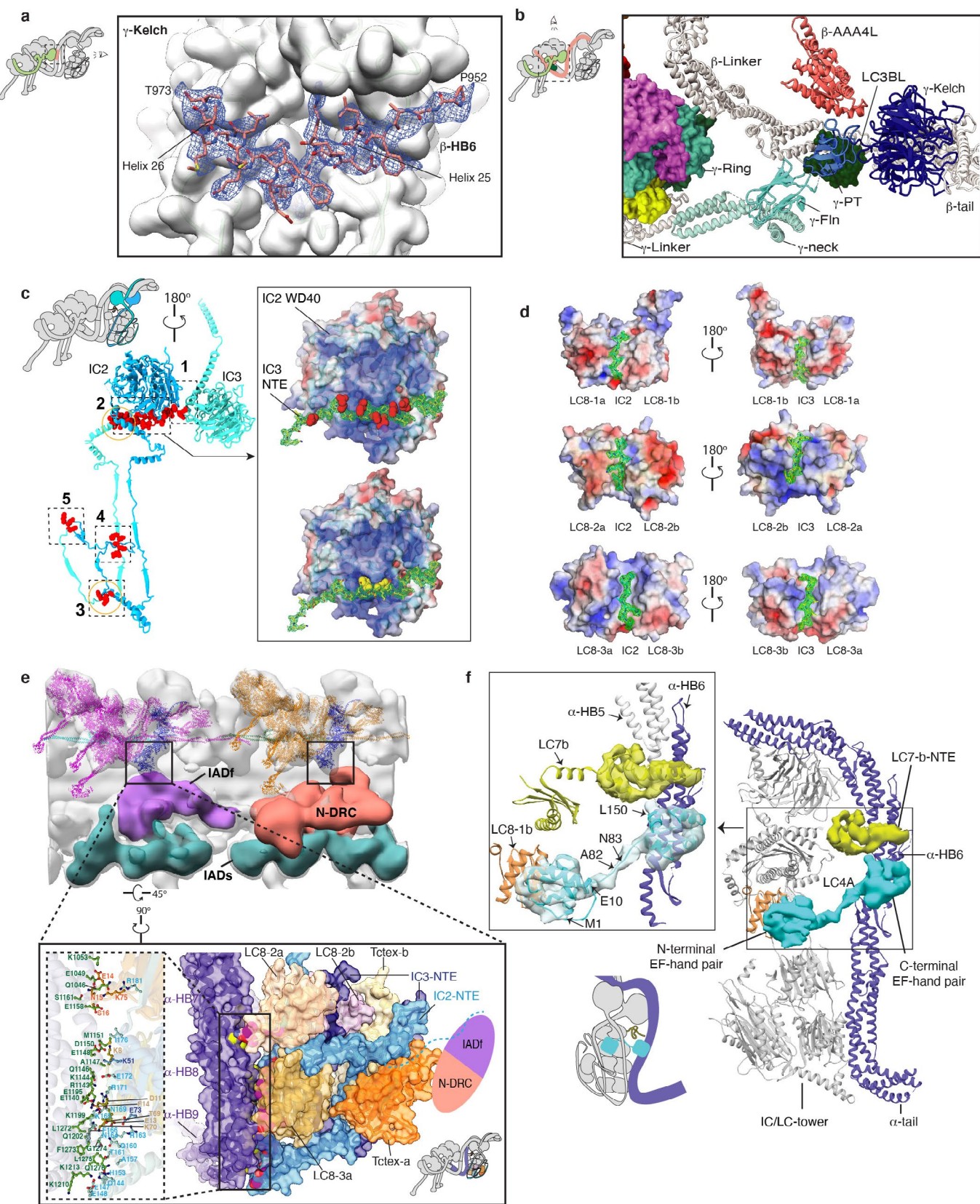

**Extended Data Fig. 5 | See next page for caption.**

**Extended Data Fig. 5 | The structure of γ-HC and IC/LC-tower. (a)** The γ-Kelch and β-tail are tightly bound together via a snug insertion of a 'V-shaped region' of β-tail into the γ-Kelch groove. **(b)** The γ-Ring is pinched between β- and γ-Linker. LC3BL resides at the joint region of β-Linker and β-neck. It links the β-Linker/neck region to γ-tail to form a locally remodelable interaction network. **(c)** IC2 and IC3 interact with each other to form an assembly scaffold. The right panel shows that the charge-charge interaction occurs between the positively charged groove (surface-charge rendering) of the IC2-WD40 domain and the negatively charged loop of IC3-NTE (residues Q125-N144), of which the negatively charged residues are shown as red sphere (upper right). In addition, two aromatic residues, Y132 and F133 (yellow sphere) of IC3-NTE insert into the small pits on the surface of IC2-WD40 (lower right). **(d)** The detailed interaction interfaces between IC2/3 and LC8s. **(e)** The Tctex-a/b and IC2/3 NTEs of the IC/LC-tower are connected to the IC-LC complex of IADf (purple volume) and N-DRC (orange volume). The enlarged view of this linking region shows that the Tctex heterodimer is backfolded with a beta-hairpin of IC2-NTE specifically snug into the groove formed by LC8-2a/b and LC8-3a (lower right inset). The IC2-NTE (residues L61-W117), IC3-NTE (residues K12-K67), LC8-2a/b and LC8-3a/b, and Tctex-a/b formed the bottom region of the IC/LC-tower, tightly bind to α-neck. The close-up view of the closely packed interactions between the bottom region of the IC/LC-tower and α-neck (left inset). The interacting residues are shown as sticks. The cyan dashed line indicates the untraced residues of the IC2. **(f)** A complete cartoon model of IC/LC-tower and α-tail, including the LC7-b-NTE and LC4A. The N-terminal EF-hand pair of LC4A binds to HB6 of the α-tail, which agrees very well with the IQ binding site[73]. The C-terminal EF-hand pair of LC4A interacts with LC8-1b.

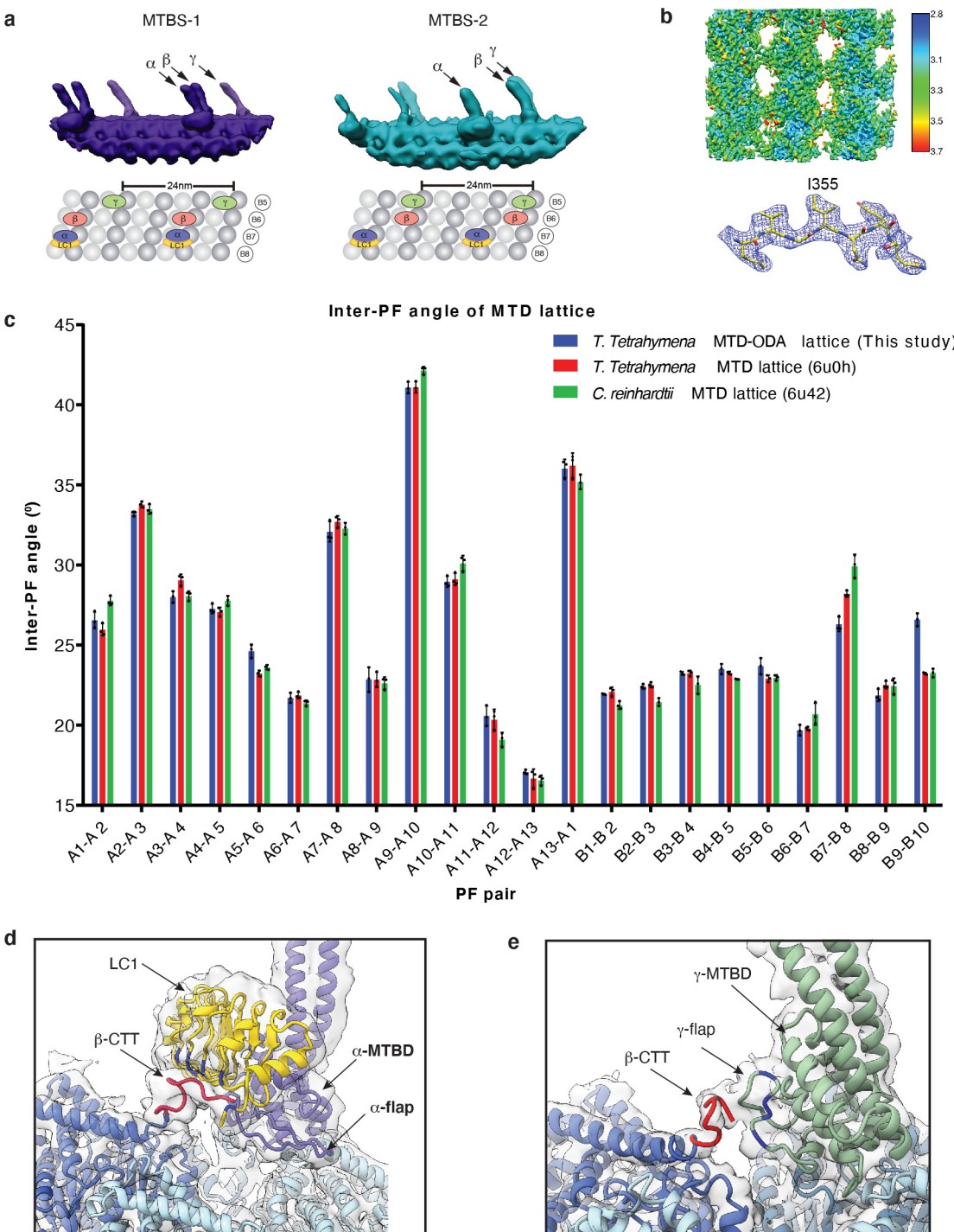

**Extended Data Fig. 6 | The structures of OAD MTBDs and their interactions with microtubule doublet. (a)** Synchronized positions of the OAD MTBDs bound to four microtubule protofilaments in MTBS1 and MTBS-2 (upper). The corresponding protofilaments of the native MTD are labeled in the attached model (lower). **(b)** The structure of MTD tubulins (16-nm repeat) locally refined to 3.08 Å resolution. **(c)** The inter-PF angles of the MTD lattice in this study show the same pattern as that of previously published ones in both *T. thermophila* (PDB: 6U0H)[36] and *C. reinhardtii* (PDB: 6U42)[15]. Each inter-PF angle was calculated three times. Data are presented as mean values ± SD. **(d-e)** The local density map of α-MTBD/LC1/tubulins complex was lowpass filtered to 6 Å to clearly show the density connection between β-CTT with LC1 **(d)** and γ-flap **(e)**.

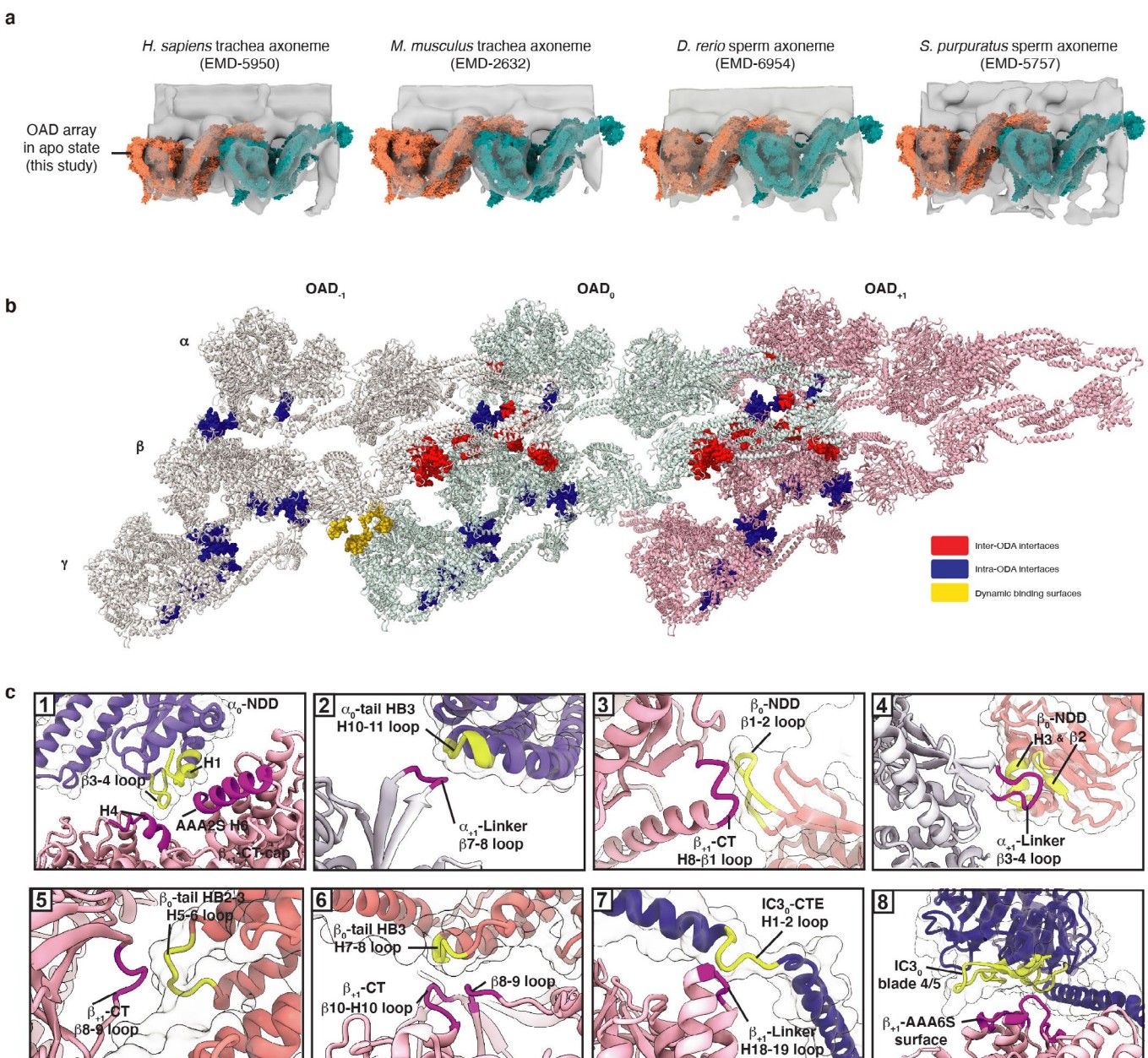

**Extended Data Fig. 7 | The conserved TTH interactions in motile cilia across species for array formation. (a)** The atomic model of our OAD array (γ-HC removed) fits well into previously reported cryo-ET maps of axoneme structures from four different organisms [human (Homo. sapiens), mouse (Mus. musculus), zebra fish (Danio. rerio), and sea urchin (Strongylocentrotus. purpuratus)][12,13,40,74], whose OADs contain homologues of α- and β-heavy chains. **(b)** The top-view structure of the array networks formed by intra- (blue) and inter-OAD (red, yellow) interactions. **(c)** Enlarged views of the eight main TTH interaction sites. The regions of $OAD_0$ and $OAD_{+1}$ involved in interactions are marked in yellow and purple, respectively.

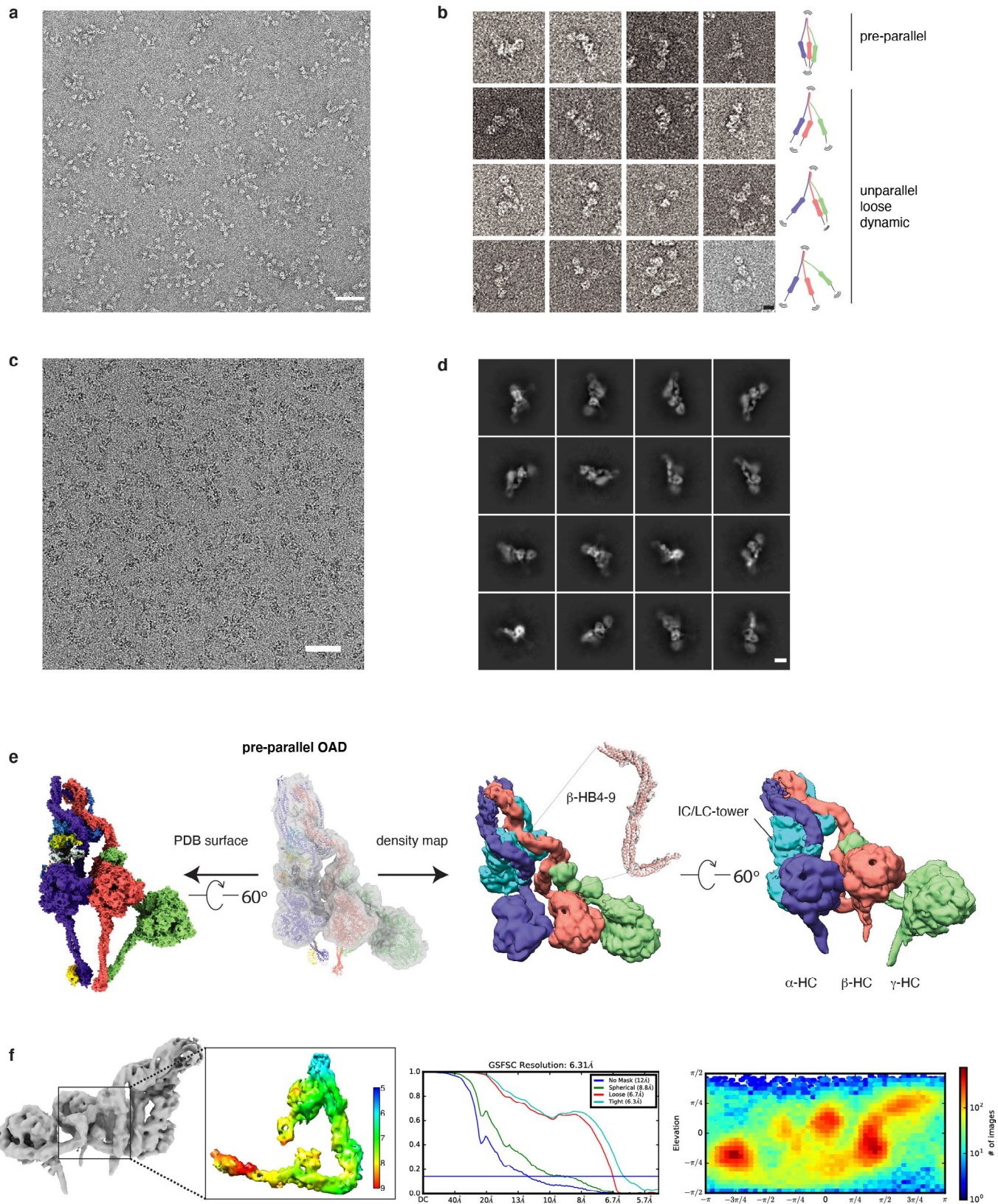

**Extended Data Fig. 8 | See next page for caption.**

**Extended Data Fig. 8 | The structure of free OAD in pre-parallel state. (a)** The representative negative staining image of the free OADs. Similar images could be regularly acquired from n > 5 independent OAD purifications. Scale bar, 100 nm (**b**) Representative two-dimensional (2D) projection of the free OAD sample with a model attached for each row. The three motor domains and tail region have a wide range of dynamic conformations. Scale bar, 10 nm. (**c**) The representative cryo-EM image of the free OADs. Similar images could be regularly acquired from n > 5 independent OAD purifications. Scale bar, 100 nm. (**d**) Representative 2D averages of the free OAD sample purified via Grafix[57]. Scale bar, 10 nm. (**e**) The cryo-EM structure of pre-parallel OAD. The atomic model of parallel OAD is fitted into the density map of pre-parallel OAD (transparency grey, upper left) as a rigid body. The cryo-EM density map of pre-parallel OAD is presented at the right panel from two different views. The local density map of the β-HB4-9 is attached. (**f**) The local resolution, FSC curve, and orientational distribution for the representative region from the free OAD.

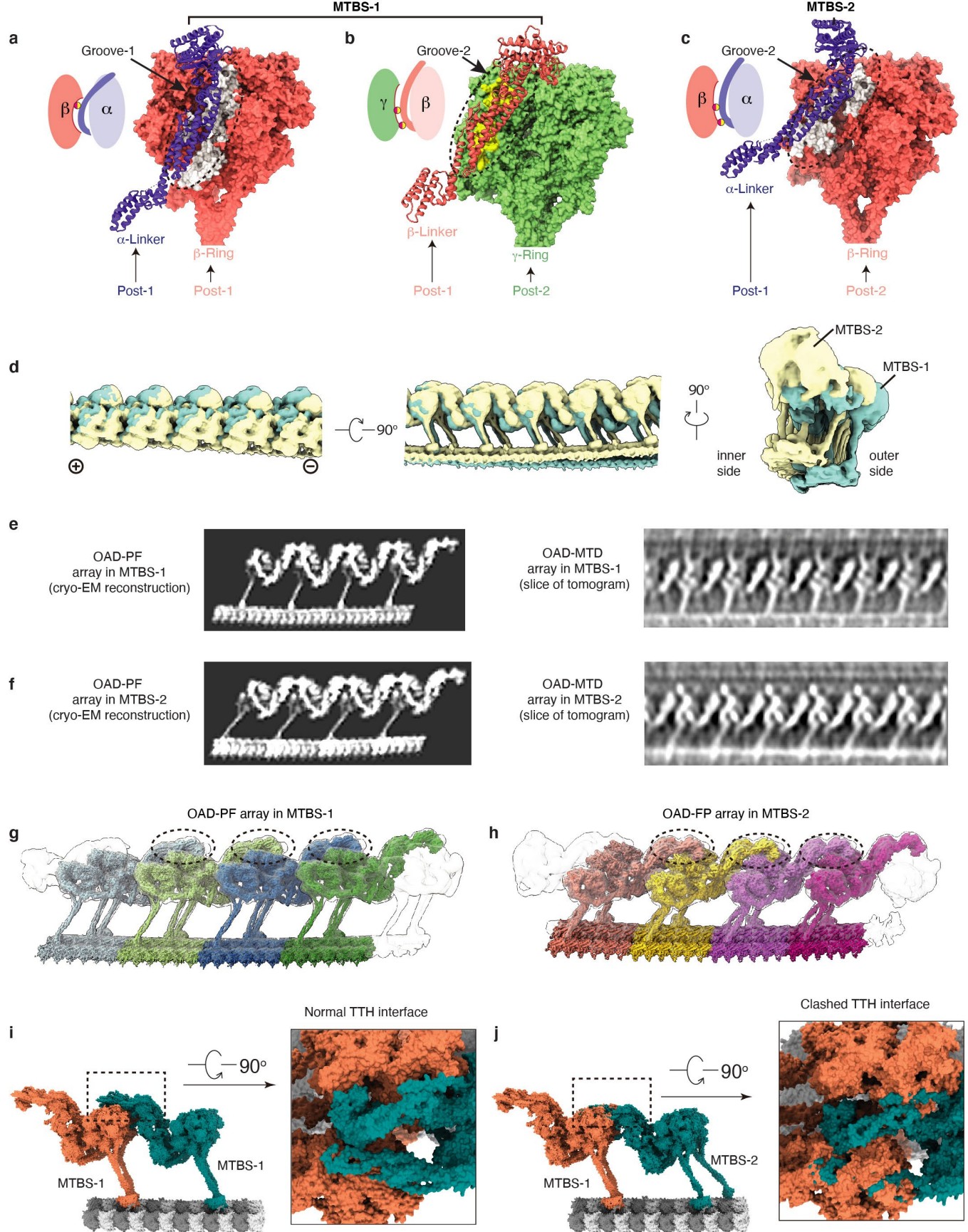

**Extended Data Fig. 9 | See next page for caption.**

**Extended Data Fig. 9 | A comparison between OAD-PF structures in MTBS-1 and MTBS-2. (a-c)** The α-Linker moves its docking site on β-Ring from Groove-1 in MTBS-1 **(a)** to Groove-2 in MTBS-2 **(c)**, which is consistent with the groove where β-Linker docks on γ-Ring in MTBS-1 **(b)**. **(d)** A comparison between OAD-PF arrays in MTBS-1 and MTBS-2 from three different views. Each array is generated by elongating the identical 48-nm OAD-PF maps in the same MTBS. **(e-f)** Cryo-EM reconstruction of the reconstituted OAD arrays (left) and the corresponding states from our cryo-ET analysis (right). The slices are focused on β-HC to show the key difference between MTBS-1 **(e)** and MTBS-2 **(f)**. The single particle reconstructions were generated by joining two identical 48-nm OAD-PF maps. The cryo-ET maps were generated by sub-volume averaging from two representative tomograms. **(g-h)** The OAD arrays in MTBS-1 and MTBS-2 are assembled in the same TTH manner. The TTH interfaces are marked by the dashed black circles. **(i-j)** Substituting one OAD unit in MTBS-1 **(i)** with a unit in MTBS-2 **(j)** severely disrupts the interactions between the IC-NDD$_0$ and the α/β$_{+1}$-LR region (**j**, right).

*M. musculus* trachea axoneme in ATP state (EMD-2633)

*S. purpuratus* sperm axoneme in ATP state (EMD-5758)

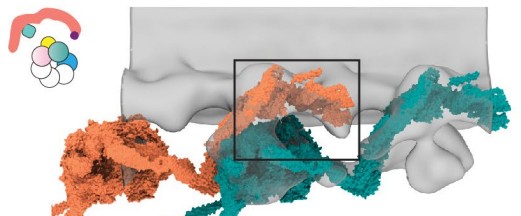 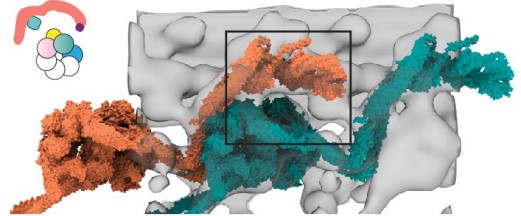

**Extended Data Fig. 10 | TTH interfaces of OAD arrays are disrupted by ATP hydrolysis.** Conformational changes of OAD array in ATP state. The models (tomato and teal surfaces) were built based on previously reported cryo-ET maps[12,40] (transparent grey) and our OAD coordinates. The TTH interfaces in ATP state are highlighted in the black squares, each illustrated with a cartoon model on the left.

# nature research

# Reporting Summary

Nature Research wishes to improve the reproducibility of the work that we publish. This form provides structure for consistency and transparency in reporting. For further information on Nature Research policies, see our Editorial Policies and the Editorial Policy Checklist.

## Statistics

For all statistical analyses, confirm that the following items are present in the figure legend, table legend, main text, or Methods section.

| n/a | Confirmed | |
|---|---|---|
| ☐ | ☒ | The exact sample size (*n*) for each experimental group/condition, given as a discrete number and unit of measurement |
| ☐ | ☒ | A statement on whether measurements were taken from distinct samples or whether the same sample was measured repeatedly |
| ☐ | ☒ | The statistical test(s) used AND whether they are one- or two-sided<br>*Only common tests should be described solely by name; describe more complex techniques in the Methods section.* |
| ☒ | ☐ | A description of all covariates tested |
| ☒ | ☐ | A description of any assumptions or corrections, such as tests of normality and adjustment for multiple comparisons |
| ☐ | ☒ | A full description of the statistical parameters including central tendency (e.g. means) or other basic estimates (e.g. regression coefficient) AND variation (e.g. standard deviation) or associated estimates of uncertainty (e.g. confidence intervals) |
| ☐ | ☒ | For null hypothesis testing, the test statistic (e.g. *F*, *t*, *r*) with confidence intervals, effect sizes, degrees of freedom and *P* value noted<br>*Give P values as exact values whenever suitable.* |
| ☒ | ☐ | For Bayesian analysis, information on the choice of priors and Markov chain Monte Carlo settings |
| ☒ | ☐ | For hierarchical and complex designs, identification of the appropriate level for tests and full reporting of outcomes |
| ☒ | ☐ | Estimates of effect sizes (e.g. Cohen's *d*, Pearson's *r*), indicating how they were calculated |

*Our web collection on statistics for biologists contains articles on many of the points above.*

## Software and code

Policy information about availability of computer code

| Data collection | SerialEM 3.7.8 |
|---|---|
| Data analysis | Relion 3.0, MotionCor 2, Gctf 1.08, Gautomatch 0.56, Cryosparc 2.12, Coot 0.8.9, Molprobity 4.5, PyMOL 2.3.2, Chimera 1.12, ChimeraX 0.6, Fiji, Refmac 5.7, IMOD 4.9.12, FIESTA 1.6.0 |

For manuscripts utilizing custom algorithms or software that are central to the research but not yet described in published literature, software must be made available to editors and reviewers. We strongly encourage code deposition in a community repository (e.g. GitHub). See the Nature Research guidelines for submitting code & software for further information.

## Data

Policy information about availability of data

All manuscripts must include a data availability statement. This statement should provide the following information, where applicable:
- Accession codes, unique identifiers, or web links for publicly available datasets
- A list of figures that have associated raw data
- A description of any restrictions on data availability

The coordinates are deposited in the Protein Data Bank with PDB IDs 7K58 (OAD-MTD in MTBS-1), 7K5B (OAD-MTD in MTBS-2), 7KEK (free OAD in pre-parallel conformation), 7N32 (four PFs of OAD-MTD), 7MWG (16-nm MTD), respectively. The cryo-EM maps are deposited in the Electron Microscopy Data Bank with IDs EMD-22677 (OAD-MTD in MTBS-1), EMD-22679 (OAD-MTD in MTBS-2), EMD-22840 (free OAD in pre-parallel conformation), EMD-24066 (16-nm MTD). Source data are provided with this paper as well.

# Field-specific reporting

Please select the one below that is the best fit for your research. If you are not sure, read the appropriate sections before making your selection.

☒ Life sciences          ☐ Behavioural & social sciences          ☐ Ecological, evolutionary & environmental sciences

For a reference copy of the document with all sections, see nature.com/documents/nr-reporting-summary-flat.pdf

# Life sciences study design

All studies must disclose on these points even when the disclosure is negative.

| | |
|---|---|
| Sample size | More than 50 microtubules were randomly selected for velocity analysis in the gliding assay. More than 15,000 cryo-EM movies were collected to achieve near-atomic reconstructions. 268, 712 high-quality particles were included in the final reconstruction.  For the inter-PF angle calculation, all reconstructed tubulins were taken into consideration. |
| Data exclusions | Electron microscopy: All micrographs were checked manually, and bad ones were excluded.<br>Microtubule gliding assay: Tracking results were manually inspected to exclude immobile filaments, surface dirt particles, tracks less than 1 second, and tracking errors due to filament collisions.<br>Other: No data was excluded from the analysis. |
| Replication | OAD purifications were repeated over 5 times. The microtubule gliding assays were repeated three times using more than 50 microtubules to estimate the velocities. The OAD array reconstitution assays were repeated for more than 5 times. Cryo-EM structures were lowpass filtered and re-refined, which generated consistent reconstructions. |
| Randomization | Microtubules were randomly selected from different areas of microscopy slides without preference to estimate the gliding velocity. |
| Blinding | Blinding was not relevant to this study. We cannot be blinded as microtubules and OAD molecules have to be identified based on experience and prior knowledge of the field. |

# Reporting for specific materials, systems and methods

We require information from authors about some types of materials, experimental systems and methods used in many studies. Here, indicate whether each material, system or method listed is relevant to your study. If you are not sure if a list item applies to your research, read the appropriate section before selecting a response.

### Materials & experimental systems

| n/a | Involved in the study |
|---|---|
| ☒ | ☐ Antibodies |
| ☐ | ☒ Eukaryotic cell lines |
| ☒ | ☐ Palaeontology and archaeology |
| ☒ | ☐ Animals and other organisms |
| ☒ | ☐ Human research participants |
| ☒ | ☐ Clinical data |
| ☒ | ☐ Dual use research of concern |

### Methods

| n/a | Involved in the study |
|---|---|
| ☒ | ☐ ChIP-seq |
| ☒ | ☐ Flow cytometry |
| ☒ | ☐ MRI-based neuroimaging |

## Eukaryotic cell lines

Policy information about cell lines

| | |
|---|---|
| Cell line source(s) | Tetrahymena Thermophila SB715 |
| Authentication | None of cell lines were authenticated. |
| Mycoplasma contamination | Mycoplasma contaminaiton was not performed in this study. |
| Commonly misidentified lines<br>(See ICLAC register) | No commonly misidentified lines were used in our study. |

