## [Peer Review File · Nature Structural & Molecular Biology]

Peer Review Information

Journal: Nature Structural and Molecular Biology

Manuscript Title: Structures of outer-arm dynein array on microtubule doublet reveal a motor coordination mechanism

Corresponding author name(s): Kai Zhang

Editorial Notes:

**Redactions –
unpublished data**

Reviewer Comments & Decisions:

Decision Letter, initial version:

17th Mar 2021

Dear Jack,

Thank you again for submitting your manuscript "Cryo-EM structures of outer-arm dynein array bound to microtubule doublet reveal a mechanism for motor coordination". I apologize for the delay in responding, which resulted from the difficulty in obtaining suitable referee reports. Nevertheless, we now have comments (below) from the 3 reviewers who evaluated your paper. In light of those glowing reports, we remain decidedly interested in your study and would like to see your response to the comments of the referees, in the form of a revised manuscript.

You will see that all reviewers were very positive about the interest and the quality of the work. Reviewer 1, an expert on dynein, was particularly enthusiastic, but expressed concerns about the readability of the paper that should be addressed by following their suggestions regarding rewriting and focusing on central aspects of the work (editorially, we cannot encourage presenting the work in two papers as suggested). Reviewer 2, an expert on cilia and cryo-EM, finds the model presented in Figure 7 too speculative and has some recommendations for revising it. Reviewer 3, also an expert on cilia and cryo-EM, was concerned about the availability of detailed information on structural data and methodology. The reviewer also requests to include a comparison of the results from computational simulations of the effects of ATPyS on the length of the OAD array with experimental data.

Please be sure to respond to all concerns of the referees in full in a point-by-point response and highlight all changes in the revised manuscript text file. If you have comments that are intended for editors only, please include those in a separate cover letter.

The revision should follow our Article format (<https://www.nature.com/nsmb/about/content#article>):

- title: 100 characters INCLUDING spaces.
- abstract: 150 words, no references; state species in which work is done
- main text: usually capped at 4,000 words, divided into introduction (typically 700 words); results (with subheadings); discussion (700 words).
- figures: we allow up to 7 display items and up to 10 Extended Data Figures
- online methods: please stay below ~3,500 words; detailed procedures can be outsourced to Supplemental Notes.
- Supplementary items: up to 10 Supplementary figures; other allowed items are Supplementary Tables, Videos, Notes.
- Extended Data Table 1 should be included in the main text as Table 1.

We expect to see your revised manuscript within 6 weeks. If you cannot send it within this time, please contact us to discuss an extension; we would still consider your revision, provided that no similar work has been accepted for publication at NSMB or published elsewhere.

Reporting Summary:

Data availability: this journal strongly supports public availability of data. All data used in accepted papers should be available via a public data repository, or alternatively, as Supplementary Information. If data can only be shared on request, please explain why in your Data Availability Statement, and also in the correspondence with your editor. Please note that for some data types, deposition in a public repository is mandatory - more information on our data deposition policies and available repositories can be found below:

<https://www.nature.com/nature-research/editorial-policies/reporting-standards#availability-of-data>

[REDACTED]

Kind regards,
Florian

Florian Ullrich, Ph.D.
Associate Editor
Nature Structural & Molecular Biology
ORCID 0000-0002-1153-2040

Reviewers' Comments:

Reviewer #1:

Remarks to the Author:

This manuscript reconstituted Tetrahymena outer arm dynein (OAD) and managed formed arrays of OADs on microtubule doublets. Using cryoEM, they were able to obtain near-atomic structure of the entire complex and discovered the stoichiometry and orientation of more than 10 light chains. They also revealed the contacts between individual motor complexes on the array under different nucleotide conditions. At the conclusion of this work, they propose a new model, in which motors on the array sequentially hydrolyze ATP and pull on the microtubule, and powerstroke of the motors bend the microtubule and allow the motors to reform the conformation for the new nucleotide cycle.

This is one of the most exciting manuscripts I have read for many years. In this study, the authors performed a highly impressive state-of-the-art cryoEM imaging and employed various data analysis approaches to address several open questions in the cilia field. The work has also produced new testable hypotheses in the field. Therefore, I wholeheartedly support the publication of this work in NSMB. I only have minor suggestions:

1. A lot is going on here (I think it would have been a great idea to split the manuscripts into two stories and publish them back-to-back in the same journal). It is not necessarily easy to follow the details of this outstanding work for someone who is not an expert in structural biology in such a dense format. I would recommend the authors to take their time and better explain some of the most fundamental findings, and perhaps omit some other details to supplement or for another paper.
2. It is not clear how the authors set up their experiments. How did they express OAD, purify microtubule doublets, form OAD arrays on the microtubule? While most of these are explained well in the Methods section, it would be great if the authors could have a paragraph briefly describing the reconstitution of the system at the beginning of the Results section.
3. Do free OADs in Figure 4c contains all of the light chains and intermediate chains? Do they only lack microtubule doublets?
4. I understand the authors' effort to introduce the concept of MT curvature as part of the OAD mechanochemical cycle shown in Figure 7. However, it is not clear why curvature is needed for the

model to work, and it gives the false impression that the authors try to hit all the benchmarks in a single story. This story is definitely not thin on data and has many remarkable findings, and it would be a better use of available space to focus on those results rather than mixing them with other untested ideas. Because the manuscript does not contain any data on microtubule curvature (or I missed it altogether), I would deemphasize this point and mention only that curvature may also play a role in the cycle.

5. The majority of the Discussion focuses too heavily on the plausible model shown in Figure 7. However, there is no discussion of other possibilities or mention of previous models proposed in the field.

6. The authors compare their MTBDs to that shown for an IAD by the Carter group (3ERR). That work proposed that the flap contacts the neighboring protofilament and distorts the MT. If I understand correctly, this is not what the authors of this study observe. Instead, they show that the flap of some MTBDs contact the C-terminal tail of beta-tubulin. This contrast should be better emphasized in the manuscript.

7. Page 10, "OADs utilize the tension for MTD bending": I would replace "bending" with "sliding" as OADs slide the MTs and flexible linker between MTs cause them to bend instead of sliding disintegration.

8. Page 10, "the motor domains function coordinately": Do the authors mean that the heavy chains take coordinated steps? This strongly contrasts the stepping studies on cytoplasmic dynein by Vale, Yildiz, and Reck-Peterson groups. There is also stepping data for OADs that show similar variability in the size and direction of steps taken by OAD. It is hard to imagine how the heads would coordinate their stepping while taking such variable steps. I would soften the language here as the manuscript does not present any direct evidence that distinguishes between coordinated vs uncoordinated stepping of the heavy chains.

9. Similarly, Page 11, "regulating the affinity between MTD and alpha-MTBD in a curvature dependent manner... is likely to be a universal mechanism": The authors do not have any data to make such claim, which seems to come out of nowhere. There is no direct evidence suggesting that LC1-betaCTT is the curvature sensor. While these points could be food for thought in a commentary or a perspective article, it is too much speculation for a research article that already covers too many aspects of OAD structure and regulation. I would recommend the authors focus their discussion on their findings and compare their model to the previous models in the field.

10. Figure 3 and other figures: I am somewhat surprised that the authors did not mention any specific residues when they discuss interactions between different parts of this assembly with each other, between neighboring OADs, and between OAD and the MT. Is this because of the lack of space in the manuscript (see my first suggestion 1 above) or because of the lack of resolution?

11. Abstract: "...illuminate how OADs coordinated with each other to move one step forward" should be deleted from this sentence as the first part of the sentence is an experimental observation, whereas this second part is an interpretation or a proposed model. I would add something like this to the end of the abstract "we propose that OADs in an array sequentially hydrolyze ATP to slide the MT doublet..."

12. Abstract, next sentence: It is not clear what the authors mean by "native-like pattern for its distinct microtubule-binding domains".

13. Introduction, second sentence: Replace "defects of motile cilia" with "defects in ciliary structure or function". Motile cilia need to be better defined in the introduction.

14. Linker and ring should start with a lower case as they are not special names.

15. Introduction, the last two sentences of the second paragraph: "Current understanding (of what)" and replace "Nevertheless, it is largely unclear" with "However, it remains unclear"

16. Page 3, "extensive network (of) interactions"

17. Page 3, it is not clear what the authors mean by "it is the ATP hydrolysis that temporarily relaxes the TTH interfaces to allow free nucleotide cycle of downstream OADs." I would recommend replacing this sentence with something like this: "TTH interactions between neighboring OADs on the array gate the ATP hydrolysis of the motors before their minus-end directed neighbor hydrolyzes ATP and releases from the microtubule."
18. In the same paragraph, delete "most detailed" as it is subjective.
19. Page 3" what do the authors mean by "the reconstituted OAD-MTD is mediated by the MTBD"?
20. Page 5. Add space between "in" and "gamma".
21. Page 5: Proposing LC1 as a curvature sensor is too speculative for the Results section and it is not related to the results shown in this manuscript.
22. Page 11. delete "as a group".
23. Figure 1c: There is an extra tick right next to "50" in the x-axis that needs to be deleted.
24. Figure 6b: The panels are too small to see what is going on in cryoEM pictures.

Reviewer #2:

Remarks to the Author:

In this manuscript, the authors described the structure of the 3-headed outer dynein arm (ODA) purified from *Tetrahymena* and then reconstituted onto the purified doublet microtubules. The ODA stacks together allowing the author to get a cryo-EM structure. The structures by the authors are the highest resolution of ODA available. This allows the authors to completely model all the heavy chains, intermediate chains and light chains, and microtubule-binding domains of the ODA with the help of mass spectrometry data. In addition, the authors also performed structure determination of free ODA, microtubule gliding assay with different nucleotide states. Another interesting insight into the ODA exists in the two Post-1 and Post-2 states.

This work represents a major breakthrough in dynein research. This detailed model will lead to a lot more research in the field later regarding the roles of regulation of dynein activities. The work is, therefore, suitable to be published in NSMB.

The paper seems to be carefully written with a lot of details presented. The figures are very well prepared. So, I believe there will not be a lot to fix here.

My concern is rather on the model presented, which is rather speculative.

- I think the correlation bending of the doublet to the bending of the cilia is not relevant. In the cilia, in the bent region, dyneins are activated and inactivated depending on the curvature of the cilia. However, in the isolated doublet reconstitution, there is no stress to induce similar conformation (doublet is not significantly bent like the curved region and ODA is not under stress because it is not bound to a doublet).

- With that concern in mind, then the author needs to be very careful to present their model on the ATP hydrolysis cycle in Figure 7 with the introduction of Apo (Post-2) conformation.

Minor concerns:

- Another minor concern is about the presentation of ODA activation in Figure 4. It is reasonable to interpret like the authors from their own data. However, we now know that ODA is transported in an inactive stage with a molecule called Shulin (Mali et al. Science, 2021) (not cited in this paper since it only recently came out but available on Biorxiv for quite a long time before). Shulin will be then detached out of ODA. Therefore, in situ, there might not exist ODA in a dynamic unparallel state

before assembly into the active state.

- The resolution of the doublet microtubule at 16-nm might be inflated due to the treatment of the doublet like single particle analysis. Normally, in helical reconstruction, the particles must be splitted based on the filament to prevent overlapping particles to inflate the resolution. In single particles, they are splitted randomly, which can be overlapped and inflate the resolution quite significantly.

Reviewer #3:

Remarks to the Author:

In this manuscript, Rao et al. provided the detailed structures of outer arm dynein (OAD) array attached to microtubule doublet (MTD) using cryo-EM.

One of major breakthroughs is the finding of spontaneously assembled OAD array on MTD via MTBD (microtubule-binding domain). Cryo-EM structures of OAD (Mali et al., 2021) or OAD-MTD (Walton et al., 2021) were recently reported, but the interaction between MTBD and MTD was not clearly resolved. This is because OADs are attached to MTD via the docking complex (opposite side of MTBD) and MTBDs are connected through flexible stalks, which make it difficult to resolve stalk and MTBD. In this study, isolated OADs from *Tetrahymena* and MTD were mixed to generate OAD array on MTD via MTBD, which enabled authors to analyze and classify the structure of MTBD bound to MTD protofilaments.

The other major finding in this work is the importance of tail-to-head (TTH) interaction of OAD with adjacent OAD. Although inter-OAD interaction was already observed in many previous cryo-ET analyses, its function has not been validated since in native axoneme, the docking complexes align OADs. *de novo* generated OAD array on MTD in this study clearly showed the requirement of TTH interaction for OAD alignment. Moreover, two types of OAD apo-states were revealed to adopt incompatible TTH conformations, suggesting allosteric regulation of OAD states through TTH interfaces. Biochemical analyses of OAD with various ATP-analogues showed sequential detachment of OADs from MTD at the end of OAD array rather than at the middle region. Taken together these data, authors proposed a most detailed model of ATP hydrolysis cycle of OADs with coordination through TTH interactions.

Overall, the amount of data and its importance are impressive, and the reviewer strongly recommends this manuscript to be published Nature Structural & Molecular Biology because the paper is an important progress contributing to understand not only detailed structure of OAD-MTD via MTBD, but also the regulatory mechanism of OAD motor coordination on MTD.

Major comments

(1) Although the reviewer appreciates the overall quality of the analysis, the current manuscript does not contain enough information to evaluate the validity of the results.

a. Extended Data Table 1: This table is misleading, since it looks as if the MTBS-1 structure is derived from Dataset 1 and the MTBS-2 structure is derived from dataset 2 and 3. In addition, the number of raw micrographs (movies) for each dataset is missing. This table should be reorganized to be consistent with the method section. For example, MTBS-1 is from 191,776 particles and MTBS-2 is from 76,936 particles, instead of 268692 ($\sim 191,776 + 76,936 = 268,712$) particles.

b. P16~17: The reviewer recommends using a figure to explain the flow of OAD-PF structure determination. For example, showing the intermediate 3D classes (there were nine good classes and one bad class according to the method) will help the reader to better understand the analyses.

- c. Extended Data Table 1 (7K5B and 7KEK): Were these two atomic models refined?
- d. p20: "Cryo-EM maps and atomic coordinates have been deposited in the Electron Microscopy Data Bank under accession codes EMD-22677, EMD-22679, EMD-22840, EMD-XXXX and in the Protein Data Bank under accession 7K58, 7K5B, 7KEK, XXXX.". Are the authors going to deposit one more structure?
- e. The local resolutions and FSC curves of MTBS-2 (EMDB-22679) and free OAD (EMDB-22840) should be also shown.
- f. As expected from the structure, there are severe orientation preference (p 16). Therefore, orientation distribution should be shown.
- (2) p.8 "Using a titration of ATPγS, we were able to gradually shorten OAD arrays in vitro owing to the low hydrolysis rate (Fig. 6c)": Fig. 6c is critical to understand the model proposed in Fig. 7c-f. Although the comparison of experimental data and computational simulation in Fig. 6c clearly denied random-disruption model of OAD arrays, the comparison of experimental data with computational simulation of ends-disruption model of OAD arrays is required to confirm the validity of the hypothesis.

Minor comments

- (1) p21 References: the format of references is not formatted for NSMB. For example, ref 3 has volume and page before the journal name.
- (2) p14: "the grids were coated a carbon layer": "with" before a carbon layer missing?
- (3) Fig. 3a: "The relative rotations between adjacent protofilaments are illustrated with a model in the upper right.": It is not clear the white circle (microtubule protofilament?) and light blue, blue, and dark blue sticks (stalks?) are.
- (4) Fig 5c: The authors compared beta-MTBDs (high-affinity state) and 3ERR (low-affinity state). As new high-affinity structures (Nishida et al., 2020) are available, the reviewer recommend to compare these structures as well.
- (5) Extended Data Fig. 6: Both MTBS-1 and MTBS-2 structures show that the protofilaments of microtubules are slightly curved toward outside. This seems consistent with the raw image shown in Extended Data Fig 1c, where well aligned OAD are located concave side of the doublet microtubules. Is this true? If so, is the slight curvature induced by OAD or from doublet microtubule itself?

References:

- Mali, G.R., Ali, F.A., Lau, C.K., Begum, F., Boulanger, J., Howe, J.D., Chen, Z.A., Rappsilber, J., Skehel, M., Carter, A.P., 2021. Shulin packages axonemal outer dynein arms for ciliary targeting. *Science* 371, 910–916. <https://doi.org/10.1126/science.abe0526>
- Nishida, N., Komori, Y., Takarada, O., Watanabe, A., Tamura, S., Kubo, S., Shimada, I., Kikkawa, M., 2020. Structural basis for two-way communication between dynein and microtubules. *Nat. Commun.* 11, 1038. <https://doi.org/10.1038/s41467-020-14842-8>
- Walton, T., Wu, H., Brown, A., 2021. Structure of a microtubule-bound axonemal dynein. *Nat. Commun.* 12, 477 . <https://doi.org/10.1038/s41467-020-20735-7>

Author Rebuttal to Initial comments

Reviewers' Comments:

Reviewer #1:

Remarks to the Author:

This manuscript reconstituted Tetrahymena outer arm dynein (OAD) and managed formed arrays of OADs on microtubule doublets. Using cryoEM, they were able to obtain near-atomic structure of the entire complex and discovered the stoichiometry and orientation of more than 10 light chains. They also revealed the contacts between individual motor complexes on the array under different nucleotide conditions. At the conclusion of this work, they propose a new model, in which motors on the array sequentially hydrolyze ATP and pull on the microtubule, and powerstroke of the motors bend the microtubule and allow the motors to reform the conformation for the new nucleotide cycle.

This is one of the most exciting manuscripts I have read for many years. In this study, the authors performed a highly impressive state-of-the-art cryoEM imaging and employed various data analysis approaches to address several open questions in the cilia field. The work has also produced new testable hypotheses in the field. Therefore, I wholeheartedly support the publication of this work in NSMB. I only have minor suggestions:

1. A lot is going on here (I think it would have been a great idea to split the manuscripts into two stories and publish them back-to-back in the same journal). It is not necessarily easy to follow the details of this outstanding work for someone who is not an expert in structural biology in such a dense format. I would recommend the authors to take their time and better explain some of the most fundamental findings, and perhaps omit some other details to supplement or for another paper.

We agree with the reviewer's comments that the paper is currently in a bit condensed style because of the complexity of the structures. Some of the structural findings and their implications are not sufficiently discussed mostly for the word limit so that we can concentrate a bit more on the current story of inter-OAD coordination. We really appreciate the reviewer's suggestion on splitting the paper into two. However, this might not be editorially feasible, based on the editor's feedback. A most likely solution is that we will write a follow-up review paper to provide more details on the structures, structural comparisons of the whole dynein family, their interactions with microtubules and other co-factors, possible regulatory sites and the roles of their binding co-factors, mechanochemical cycles of dynein, force generation, and several other possible topics together with their implications. We do have interesting findings on quite a few other things in addition to what have been presented in the current paper. However, it is nearly impossible for us to touch all these topics at the moment, and we believe it is best for future work.

2. It is not clear how the authors set up their experiments. How did they express OAD, purify microtubule doublets, form OAD arrays on the microtubule? While most of these are explained well in the Methods

section, it would be great if the authors could have a paragraph briefly describing the reconstitution of the system at the beginning of the Results section.

We thank the reviewer's suggestion and have updated the main text to include a bit more details on sample preparation and biochemical assays. However, due to the word limit, it seems that we cannot include further details, but leave them in the supplementary materials.

3. Do free OADs in Figure 4c contains all of the light chains and intermediate chains? Do they only lack microtubule doublets?

Yes, the free OADs contain all chains, which were further confirmed by LC/MS analysis in this study. Actually, the free OAD we show here contains exactly the same components as what were used for OAD-MTD reconstitution. The reason that we only used α - and β -heavy chain (HC) here in the model was for simplicity of representation. We have removed all the rest of the structural components in the model to highlight the main differences between the pre-parallel state and fully aligned parallel structure. Especially, we removed the γ -HC because our structure suggests it is not essential for OAD array formation, which is also consistent with the conclusion on the non-essential role of α -HC for OAD assembly in *Chlamydomonas* (equivalent to γ -HC in *Tetrahymena*) from previous studies.

4. I understand the authors' effort to introduce the concept of MT curvature as part of the OAD mechanochemical cycle shown in Figure 7. However, it is not clear why curvature is needed for the model to work, and it gives the false impression that the authors try to hit all the benchmarks in a single story. This story is definitely not thin on data and has many remarkable findings, and it would be a better use of available space to focus on those results rather than mixing them with other untested ideas. Because the manuscript does not contain any data on microtubule curvature (or I missed it altogether), I would deemphasize this point and mention only that curvature may also play a role in the cycle.

We really appreciate the excellent points this reviewer has raised. Indeed, the role of MTD in dynein activity regulation is still full of mystery, despite the fact many people believe it is important for controlling dynein activity. The key question is 'how'. In the discussion section, we raised the possibility of curvature-dependent control of dynein activity, and that it is. We did not intend to claim further credit that our current paper already answered this question, but simply show the possibility. It might simply be an issue of inaccurate expression.

To make the points clear, our current understanding is that MTD curvature and dynein activity can potentially affect each other:

(A) On one hand, the microtubule curvature is the result from the ciliary beating, which is eventually generated by dynein conformational changes and its interaction with microtubule.

(B) On the other hand, the curvature can in turn control dynein-microtubule affinity and/or dynein activity, which is a long-standing hypothesis in the field.

Our model mainly tries to explain how neighboring OADs cooperatively change their conformations to potentially lead a relative sliding between two adjacent MTDs, which is eventually converted to MTD bending. We actually try to avoid discussing too much about 'how' MTD curvatures affect OAD activity, but simply raise the point. This was also the reason why we put a question mark on the speculated curvature control of dynein conformation from Post-2 to Post-1. We still do not know the answer at the current stage.

For the former point (A), our two structures of MTD-OAD in MTBS-1 and MTBS-2 combined with the previously reported cryo-ET structures could already provide a strongly support. We agree there are potentially more steps during an entire mechanochemical cycle of OAD. We did not intend to claim that all these are 100% based on our own data (which is impossible). Indeed, it has taken non-trivial efforts from many labs to reveal different dynein structures by various approaches in the past decades and we clearly cited those important findings that finally led us to weave together different functional states of dynein. Currently, quite a few different models have been proposed to explain dynein functional cycles. Each emphasizes different aspects but may have more or less controversial explanations compared to another. However, there is one agreement that the cilium field has reached for long: the bending of cilia is generated by sliding of adjacent MTDs, attributed to the movement of dynein. Our current model explains how dynein takes one step on microtubule. We believe the model is promising because it reconciles quite a few previously controversial explanations and agrees with most structural observations reported to date (we compared nearly all available dynein structures).

For the latter (B), we only proposed a possible role of MTD curvature in regulating conformational changes from Post-2 to Post-1. It was simply our hypothesis that the MTD curvature affects the conformational changes from Post-2 to Post-1, by a combination of our structure with previous studies from other labs. We did not intend to answer all questions in one paper, but merely to raise the possibility that MTD curvature may play a role in regulating OAD cycle. We do have some preliminary evidence in the lab that the microtubule curvature indeed affects the binding affinity of OADs, but this is not possible to be included and further discussed in one paper as the paper is already a bit too dense. To dispel the confusion, we have removed our speculation that OADs move back from Post-2 to Post-1 in a curvature dependent manner, but simply leave it as an open question.

5. The majority of the Discussion focuses too heavily on the plausible model shown in Figure 7. However, there is no discussion of other possibilities or mention of previous models proposed in the field.

We have included a comparison with models proposed by other labs in the discussion as well as previous studies that led us to propose the current model in combination with our own results.

6. The authors compare their MTBDs to that shown for an IAD by the Carter group (3ERR). That work proposed that the flap contacts the neighboring protofilament and distorts the MT. If I understand correctly, this is not what the authors of this study observe. Instead, they show that the flap of some MTBDs contact the C-terminal tail of beta-tubulin. This contrast should be better emphasized in the manuscript.

The 3ERR is the model of cytoplasmic dynein MTBD, which has a very short flap segment. The MTBD structure that this reviewer mentioned is actually 6RZA determined by the Carter group in a recent eLife paper rather than 3ERR. We have added a comparison between our structure and 6RZA in the revised manuscript and briefly mentioned the difference. There are at least two possibilities that we did not see the distortion as observed in the eLife paper.

First, the Carter group used singlet microtubules instead of microtubule doublets (MTDs) in their experiment. The MTD contains a network of microtubule inner proteins (MIPs) that probably impose stress on the microtubules and restrict the relative rotations between adjacent protofilaments.

Second, IAD and OAD may impose different forces on microtubules. Our current understanding is that OAD is the main force-generator, while IAD plays an important role in beating regulation. The microtubule distortion apparently requires lots of energy. It does not look like an energetically efficient system if OADs consume too much energy to simply distort MTD. Low-efficient systems must have already been avoided during evolution. However, IAD is different, as their role is to regulate the ciliary beating. A distortion among MTD protofilaments by IAD might be critically important for controlling the OAD activities during the mechanochemical cycle, in addition the OAD activity regulation by MTD curvature changes. Therefore, we believe such difference makes prominent sense based on their differentiated roles in ciliary beating. However, as our manuscript is already over the word limit, we do not think it is possible to discuss such details in the current main text.

7. Page 10, “OADs utilize the tension for MTD bending”: I would replace “bending” with “sliding” as OADs slide the MTs and flexible linker between MTs cause them to bend instead of sliding disintegration.

We appreciate the reviewer’s suggestion and have modified the description in the text.

8. Page 10, “the motor domains function coordinately”: Do the authors mean that the heavy chains take coordinated steps? This strongly contrasts the stepping studies on cytoplasmic dynein by Vale, Yildiz, and Reck-Peterson groups. There is also stepping data for OADs that show similar variability in the size and direction of steps taken by OAD. It is hard to imagine how the heads would coordinate their stepping while taking such variable steps. I would soften the language here as the manuscript does not present any direct evidence that distinguishes between coordinated vs uncoordinated stepping of the heavy chains.

There might have been some inaccurate expression in the previous manuscript. We intended to emphasize the coordination between neighboring OADs due to the TTH interfaces. The three heavy chains within the OAD may not need strict coordination in the free form as showed by previous studies on stepping behaviors of OADs. Our own cryo-EM classification also suggested all three motor domains can move freely, which agrees with previous conclusion. Our speculation is that when they form a compact array, the free motion that exists on individual OADs may be restricted in the form of an array because the three motor domains are stacked together and restricted by the tail of an adjacent OAD. To eliminate this confusion, we have rewritten this section and made the description clearer.

9. Similarly, Page 11, “regulating the affinity between MTD and alpha-MTBD in a curvature dependent manner... is likely to be a universal mechanism”: The authors do not have any data to make such claim, which seems to come out of nowhere. There is no direct evidence suggesting that LC1-betaCTT is the curvature sensor. While these points could be food for thought in a commentary or a perspective article, it is too much speculation for a research article that already covers too many aspects of OAD structure and regulation. I would recommend the authors focus their discussion on their findings and compare their model to the previous models in the field.

First, we have to apologize that we had missed some citations here. The curvature-controlling role of LC1 was previously proposed by other studies, not by us for the first time, for example:

1. Toda, A., Nishikawa, Y., Tanaka, H., Yagi, T. & Kurisu, G. The complex of outer-arm dynein light chain-1 and the microtubule-binding domain of the gamma heavy chain shows how axonemal dynein tunes ciliary beating. *J Biol Chem* 295, 3982-3989 (2020).

2. Patel-King, R.S. & King, S.M. An outer arm dynein light chain acts in a conformational switch for flagellar motility. *J Cell Biol* 186, 283-95 (2009).

Our structure shows that the β -CTT from adjacent MT is attached to LC1 and provides a direct evidence in structure that LC1 may regulate OAD activity through its interaction with CTT. This interaction can be potentially altered during ciliary bending as the relative distance between CTT and LC1 will change. On

the other hand, the strength of charge-charge interaction is very sensitive to changes of distance. Eventually, the MT-affinity of alpha-MTBD is altered. Therefore, we believe the previously proposed curvature-controlling model makes sense and is strongly supported by our current structural observation. We have cited previous literature and updated the text.

10. Figure 3 and other figures: I am somewhat surprised that the authors did not mention any specific residues when they discuss interactions between different parts of this assembly with each other, between neighboring OADs, and between OAD and the MT. Is this because of the lack of space in the manuscript (see my first suggestion 1 above) or because of the lack of resolution?

For both reasons. First of all, the resolutions of microtubules and most regions of OAD, including MTBD regions, are good enough to allow us to see side chains clearly. However, as the main point of the manuscript was to understand the overall behavior of arrayed OADs and their interaction with microtubules, we had deliberately tried to avoid dipping into too many structural details so that we could focus a bit more on the key message. The manuscript could have easily been bloated if we had discussed lots of residue-residue interactions.

On the other hand, there are regions that are indeed quite flexible, such as the CTT and N-terminal tail. We could assign the location of β -CTT density after low-pass, but the resolution was not good enough to allow precise assignment of residues. This allowed us to directly visualize the location of tubulin CTT and its interaction with dynein. We suspect that a more detailed structure of the dynamic CTT with side chain information might not be possible by currently available techniques, unless it is substantially stabilized in a fixed conformation by some microtubule-associated protein. As shown in the Extended Data Figure 4c, we could visualize good details of the interface between adjacent OADs, but it was not easy to determine the key residues, because there are too many involved. Also, the resolutions in different tail regions vary a lot. We could only build a backbone model for some regions near NDD. We did analyze the structure carefully and even built a homologous model of human OAD to figure out some disease-related mutations that potentially affect the TTH interaction and lead to PCD disease in human. However, due to the resolution limit, we would rather not take the risk of over-claiming their implications at the current.

11. Abstract: "...illuminate how OADs coordinated with each other to move one step forward" should be deleted from this sentence as the first part of the sentence is an experimental observation, whereas this second part is an interpretation or a proposed model. I would add something like this to the end of the abstract "we propose that OADs in an array sequentially hydrolyze ATP to slide the MT doublet..."

We appreciate the reviewer's suggestion and have improved the text.

12. Abstract, next sentence: It is not clear what the authors mean by “native-like pattern for its distinct microtubule-binding domains”.

We appreciate the reviewer's suggestion and have improved the text.

13. Introduction, second sentence: Replace “defects of motile cilia” with “defects in ciliary structure or function”. Motile cilia need to be better defined in the introduction.

We appreciate the reviewer's suggestion and have improved the text.

14. Linker and ring should start with a lower case as they are not special names.

We appreciate the reviewer's suggestion. Considering it is a specific region of dynein complex rather than a general 'linker', we had decided to use 'Linker' to avoid confusion with other structural features, such as the linker between the two EF-hands of calmodulins or the linker between two AAA+ domains. We find it makes the expression more specific and clearer compared to lower case as there are too many 'linkers' in a typical protein structure paper. In this case, our 'Linker' is specifically defined as helical bundles between dynein tail and AAA+ ring, while 'linker' can be flexibly used for other purposes, which we did in the paper. Also, 'Ring' is specifically defined as 'AAA+ ring' for abbreviation. This avoids the ambiguity if we talk other rings, such as 'the ring of kelch repeats' or 'the ring of WD domains'. We find it helps.

15. Introduction, the last two sentences of the second paragraph: “Current understanding (of what)” and replace “Nevertheless, it is largely unclear” with “However, it remains unclear”

We appreciate the reviewer's suggestion and have improved the text.

16. Page 3, “extensive network (of) interactions”

We appreciate the reviewer's suggestion and have improved the text.

17. Page 3, it is not clear what the authors mean by “it is the ATP hydrolysis that temporarily relaxes the TTH interfaces to allow free nucleotide cycle of downstream OADs.” I would recommend replacing this sentence with something like this: “TTH interactions between neighboring OADs on the array gate the ATP hydrolysis of the motors before their minus-end directed neighbor hydrolyzes ATP and releases from the microtubule.”

We appreciate the reviewer’s suggestion and have improved the text.

18. In the same paragraph, delete “most detailed” as it is subjective.

We appreciate the reviewer’s suggestion and have expressed it in a modest way.

19. Page 3” what do the authors mean by “the reconstituted OAD-MTD is mediated by the MTBD”?

This might be an issue of unclear expression in our manuscript. As OADs can bind to the MTD with either MTBDs or the tail region or both. Here, we were trying to emphasize that MTD interacts with the MTBD region of arrayed OADs in our reconstituted OAD-MTD. There might exist a biochemical condition that allows the tail region of OADs to attach to native MTDs in the presence of docking complex. We have updated the text to make the point clearer.

20. Page 5. Add space between “in” and “gamma”.

We appreciate the reviewer’s careful reading and have modified the text.

21. Page 5: Proposing LC1 as a curvature sensor is too speculative for the Results section and it is not related to the results shown in this manuscript.

This is the same as question 9. We have added citations and updated the text.

22. Page 11. delete “as a group”.

We appreciate the reviewer’s suggestion. It has been deleted.

23. Figure 1c: There is an extra tick right next to “50” in the x-axis that needs to be deleted.

We appreciate the reviewer’s suggestion. It has been modified.

24. Figure 6b: The panels are too small to see what is going on in cryoEM pictures.

We appreciate the reviewer’s suggestion. The views of EM images have been enlarged to show more details.

Reviewer #2:

Remarks to the Author:

In this manuscript, the authors described the structure of the 3-headed outer dynein arm (ODA) purified from *Tetrahymena* and then reconstituted onto the purified doublet microtubules. The ODA stacks together allowing the author to get a cryo-EM structure. The structures by the authors are the highest resolution of ODA available. This allows the authors to completely model all the heavy chains, intermediate chains and light chains, and microtubule-binding domains of the ODA with the help of mass spectrometry data. In addition, the authors also performed structure determination of free ODA, microtubule gliding assay with different nucleotide states. Another interesting insight into the ODA exists in the two Post-1 and Post-2 states.

This work represents a major breakthrough in dynein research. This detailed model will lead to a lot more research in the field later regarding the roles of regulation of dynein activities. The work is, therefore, suitable to be published in NSMB.

The paper seems to be carefully written with a lot of details presented. The figures are very well prepared. So, I believe there will not be a lot to fix here.

My concern is rather on the model presented, which is rather speculative.

- I think the correlation bending of the doublet to the bending of the cilia is not relevant. In the cilia, in the bent region, dyneins are activated and inactivated depending on the curvature of the cilia. However, in the isolated doublet reconstitution, there is no stress to induce similar conformation (doublet is not significantly bent like the curved region and ODA is not under stress because it is not bound to a doublet).

- With that concern in mind, then the author needs to be very careful to present their model on the ATP hydrolysis cycle in Figure 7 with the introduction of Apo (Post-2) conformation.

First of all, we feel a bit sorry that we might not fully understand the reviewer’s question (*I think the*

correlation bending of the doublet to the bending of the cilia is not relevant). Cilia is composed of microtubule doublets, dynein motors and many other complexes. Bending of cilia is eventually determined by dynein-driven MTD sliding. Cilia bending and microtubule doublet bending have to be correlated. As questioned by Reviewer 1, curvature-dependent control of dynein activity is still largely a hypothesis and needs further investigation. It was not our intention to answer how curvature affects dynein activity.

Our model in Figure 7 aims to provide an explanation on how OADs along the axoneme coordinately walk one step ahead. This is for sure to result in a MTD sliding *in vitro* and will be converted into MTD bending *in vivo*. Such coordination is controlled by TTH interactions between adjacent OADs. From Fig. 7a to Fig. 7h, our two structures of MTD-OAD in MTBS-1 and MTBS-2 and the previously reported cryo-ET structures along with our nucleotide-dependent disruption of OAD arrays at their ends supported our proposed model very well. While from Fig. 7h to Fig. 7a, we only intend to propose a possible role of MTD bending that may involve in regulating conformational changes from Post-2 to Post-1. We agree this point was indeed too speculative at the current stage. To dispel the confusion, we have removed our speculation that OADs move back from Post-2 to Post-1 in a curvature dependent manner, but simply leave it as an open question.

We agree that the conformations of arrayed OADs *in vivo* might be more complicated than what we observed *in vitro*. However, this does not mean what we observe *in vitro* will not reflect the natures of OAD. We have observed that the two microtubule-binding states, MTBS1 and MTBS2, are correlated with different MTD-bending curvatures. The curvature might not be induced by MTBS alternation, but rather that the OADs adapt their conformations to the MTD curvatures *in vitro*. This suggests that microtubule-binding states of OADs and MTD curvatures are indeed correlated. How OAD conformations and MTD curvatures affect each other in terms of precise geometrical and quantitative changes is far beyond current understanding in the field. *In vivo*, it also requires many other ciliary components to work together (central pair, radial spokes, adjacent MTDs, local chemical conditions, nucleotides, local ion strength, calcium etc.). For such level of question, we have to admit we don't know the answer yet. The main aim of our current study is simply to answer a yes-or-no question: whether OAD conformations and MTD curvatures are correlated (even if it is *in vitro* system). We did not intend to claim the differences we observe *in vitro* using biochemical and structural approaches are exactly the same as the *in vitro*. We just wanted to show these two factors are correlated, which raises the possibility that they are correlated during ciliary beating *in vivo*. We agree there are more factors involved and different tensions imposed on MTDs, but it will be really surprising if some basic properties of a protein complex observed *in vitro* do not reflect anything about its behavior *in vivo*.

The other important point is that our model is not merely based on the observation of structural differences between MTBS1 and MTBS2. We also combined current knowledge in the field and studies from other labs, particularly the previous model by J. Lin et al, 2014. We have carefully revised our figure and texts to avoid over-confidence in the proposed model.

Minor concerns:

- Another minor concern is about the presentation of ODA activation in Figure 4. It is reasonable to interpret like the authors from their own data. However, we now know that ODA is transported in an inactive stage with a molecule called Shulin (Mali et al. Science, 2021) (not cited in this paper since it only recently came out but available on Biorxiv for quite a long time before). Shulin will be then detached out of ODA. Therefore, in situ, there might not exist ODA in a dynamic unparallel state before assembly into the active state.

First of all, we really appreciate the reviewer's excellent point on Shulin. We actually determined the first structure in this manuscript more than one and half years ago [REDACTED] At the current stage, we would rather not discuss anything about Shulin as the topic coverage in our manuscript is already too dense, while Shulin is a new discovery with many interesting things to be discussed and studied.

It is still largely unknown how Shulin is really released from OAD-Shulin complex after the ciliary targeting. Even though Shulin is required for ciliary targeting, it does not rule out the possibility that it falls off OAD-Shulin complex before OAD array formation. Actually, the answer is nearly 100% sure in our own understanding of structures of dynein family, as the OAD-Shulin complex is in a phi-particle-like conformation, which severely violates the parallel conformation and tail-to-head interaction required for OAD array formation as suggested in our own study. In other words, our conclusion from structural analysis is that: Shulin must fall off OAD-Shulin complex first to allow OAD array formation, otherwise the phi-particle conformation does not allow the existence of parallel architecture. The conclusion applies to both cytoplasmic dynein-1 and OAD from currently available structures, and probably applies to dynein-2 as well if we have an open structure of dynein-2 in an array. How the OAD array is formed in the presence of Shulin and possibly other factors *in vivo* is largely unknown (and Shulin is too new in the field). Also, it is clear that Shulin knock-out does not completely remove OADs from cilia (Fig. 1D in the Shulin paper). Furthermore, there are a considerable ratio of Shulin-KO Tetrahymena cells that swim no worse than wild-type (Fig. 1C in the Shulin paper) and even faster than many wildtype cells. This implies that OAD array formation *in vivo* is indeed affected by Shulin knockout simply because the ciliary targeting of OAD is affected after Shulin KO. Without Shulin, there is still a small ratio of OADs that are transported into cilia (possibly by free diffusion) and form OAD arrays to power the ciliary beating.

Therefore, we would leave this level of discussion as open questions for future investigation and stick to our current understanding without being affected by other publications around the same time.

- The resolution of the doublet microtubule at 16-nm might be inflated due to the treatment of the doublet like single particle analysis. Normally, in helical reconstruction, the particles must be splitted based on the filament to prevent overlapping particles to inflate the resolution. In single particles, they are splitted randomly, which can be overlapped and inflate the resolution quite significantly.

We appreciate the reviewer's comments. Indeed, the overlapped signals of filamentous structures can lead to an overestimation of the resolution. To avoid such effects, we did the following tricks during cryo-EM data processing: (1) We did multiple cycles of classification. (2) We manually checked each individual filament to make sure there was no overlapping or false filamentous segments for each 16-nm repeat. (3) The final mask we imposed on local refinement did not exceed 16 nm longitudinally, which guaranteed the signals that are used for final resolution estimation are not overlapped.

Reviewer #3:

Remarks to the Author:

In this manuscript, Rao et al. provided the detailed structures of outer arm dynein (OAD) array attached to microtubule doublet (MTD) using cryo-EM.

One of major breakthroughs is the finding of spontaneously assembled OAD array on MTD via MTBD (microtubule-binding domain). Cryo-EM structures of OAD (Mali et al., 2021) or OAD-MTD (Walton et al., 2021) were recently reported, but the interaction between MTBD and MTD was not clearly resolved.

This is because OADs are attached to MTD via the docking complex (opposite side of MTBD) and MTBDs are connected through flexible stalks, which make it difficult to resolve stalk and MTBD.

In this study, isolated OADs from *Tetrahymena* and MTD were mixed to generate OAD array on MTD via MTBD, which enabled authors to analyze and classify the structure of MTBD bound to MTD protofilaments.

(Lacey et al., 2019; Nishida et al., 2020)

The other major finding in this work is the importance of tail-to-head (TTH) interaction of OAD with adjacent OAD. Although inter-OAD interaction was already observed in many previous cryo-ET analyses, its function has not been validated since in native axoneme, the docking complexes align OADs. *de novo* generated OAD array on MTD in this study clearly showed the requirement of TTH interaction for OAD alignment. Moreover, two types of OAD apo-states were revealed to adopt incompatible TTH conformations, suggesting allosteric regulation of OAD states through TTH interfaces. Biochemical analyses of OAD with various ATP-analogues showed sequential detachment of OADs from MTD at the

end of OAD array rather than at the middle region. Taken together these data, authors proposed a most detailed model of ATP hydrolysis cycle of OADs with coordination through TTH interactions. Overall, the amount of data and its importance are impressive, and the reviewer strongly recommends this manuscript to be published Nature Structural & Molecular Biology because the paper is an important progress contributing to understand not only detailed structure of OAD-MTD via MTBD, but also the regulatory mechanism of OAD motor coordination on MTD.

Major comments

(1) Although the reviewer appreciates the overall quality of the analysis, the current manuscript does not contain enough information to evaluate the validity of the results.

a. Extended Data Table 1: This table is misleading, since it looks as if the MTBS-1 structure is derived from Dataset 1 and the MTBS-2 structure is derived from dataset 2 and 3. In addition, the number of raw micrographs (movies) for each dataset is missing. This table should be reorganized to be consistent with the method section. For example, MTBS-1 is from 191,776 particles and MTBS-2 is from 76,936 particles, instead of 268692 ($\sim 191,776 + 76,936 = 268,712$) particles.

We really thank the reviewer for his rigorous reading and apologize for the mistakes we made. We have double checked and revised the data processing flow chart and updated the Extended Data Table 1.

b. P16~17: The reviewer recommends using a figure to explain the flow of OAD-PF structure determination. For example, showing the intermediate 3D classes (there were nine good classes and one bad class according to the method) will help the reader to better understand the analyses.

We appreciate the reviewer's suggestion and have added the flow chart in our revised manuscript.

c. Extended Data Table 1 (7K5B and 7KEK): Were these two atomic models refined?

We have refined the entire model of 7K5B (OAD-PF in MTBS-1). The atomic model for OAD-PF in MTBS-2 was generated based on MTBS-1. We only performed complete refinement for the three motor domains of OAD in MTBS-2 as the resolutions are better than 4 Å. For the rest of the structure, we simply divided it into different domains and extracted each of them from MTBS-1 and performed rigid-body fitting. We have updated the text and Table to make it clear.

d. p20: "Cryo-EM maps and atomic coordinates have been deposited in the Electron Microscopy Data Bank under accession codes EMD-22677, EMD-22679, EMD-22840, EMD-XXXX and in the Protein Data Bank under accession 7K58, 7K5B, 7KEK, XXXX.". Are the authors going to deposit one more structure?

Yes, we had numerous difficulties during deposition of the structures as they were too large. The codes have been updated.

e. The local resolutions and FSC curves of MTBS-2 (EMDB-22679) and free OAD (EMDB-22840) should be also shown.

We appreciate the reviewer's suggestion and have added the local resolution maps for MTBS-2 and free OAD in the Extended Data Figure 3 and Extended Data Figure 8

f. As expected from the structure, there are severe orientation preference (p 16). Therefore, orientation distribution should be shown.

Yes, the orientation preference is strong, but not at a level that theoretically disallows 3D reconstruction. Luckily, the orientation distribution was just sufficient for a reconstruction that was good enough for us to build the atomic models. We have added the orientation distribution for each local maps.

(2) p.8 "Using a titration of ATP γ S, we were able to gradually shorten OAD arrays in vitro owing to the low hydrolysis rate (Fig. 6c)": Fig. 6c is critical to understand the model proposed in Fig. 7c-f. Although the comparison of experimental data and computational simulation in Fig. 6c clearly denied random-disruption model of OAD arrays, the comparison of experimental data with computational simulation of ends-disruption model of OAD arrays is required to confirm the validity of the hypothesis.

We have added a simulation for end-release.

Minor comments

(1) p21 References: the format of references is not formatted for NSMB. For example, ref 3 has volume and page before the journal name.

We really thank the reviewer's carefulness. It was indeed our fault during the manuscript formatting for software issues. It has been double checked and corrected.

(2) p14: "the grids were coated a carbon layer": "with" before a carbon layer missing?

the grids were coated with a carbon layer

We really thank the reviewer's carefulness. We have corrected the text.

(3) Fig. 3a:"The relative rotations between adjacent protofilaments are illustrated with a model in the

upper right.”: It is not clear the white circle (microtubule protofilament?) and light blue, blue, and dark blue sticks (stalks?) are.

We have revised the model and explained more details about the model in the figure legend.

(4) Fig 5c: The authors compared beta-MTBDs (high-affinity state) and 3ERR (low-affinity state). As new high-affinity structures (Nishida et al., 2020) are available, the reviewer recommend to compare these structures as well.

We really thank the reviewer’s suggestion. It was indeed an excellent study by Nishida et al. We absolutely knew this work. It was simply an issue of timing. We actually determined the structures and started preparing the manuscript more than one year ago. Indeed, there were quite a few new findings coming out during this period, including the work by Nishida et al. It was not our intention to omit it at the beginning. We have added a comparison in the text.

(5) Extended Data Fig. 6: Both MTBS-1 and MTBS-2 structures show that the protofilaments of microtubules are slightly curved toward outside. This seems consistent with the raw image shown in Extended Data Fig 1c, where well aligned OAD are located concave side of the doublet microtubules. Is this true? If so, is the slight curvature induced by OAD or from doublet microtubule itself?

We really thank this reviewer’s expert question. This is really an excellent point. In fact, OADs can bind to MTD on both sides, concave and convex (Extended Data Fig 1c). [REDACTED] However, we tried to avoid discussing on this point as it is another level of question related to the dynamic binding affinity between OADs and MTDs during ciliary beating. We feel our current results are too limited to answer such level of questions, but it is very important for better understanding of OAD’s dynamic behavior and the mechanism of curvature-controlled OAD-MTD affinity. On the safe side and also due to the fact there are already too many points discussed in the current paper, we would rather leave it as an open question for future study.

References:

Mali, G.R., Ali, F.A., Lau, C.K., Begum, F., Boulanger, J., Howe, J.D., Chen, Z.A., Rappsilber, J., Skehel, M., Carter, A.P., 2021. Shulin packages axonemal outer dynein arms for ciliary targeting. *Science* 371, 910–916. <https://doi.org/10.1126/science.abe0526>

Nishida, N., Komori, Y., Takarada, O., Watanabe, A., Tamura, S., Kubo, S., Shimada, I., Kikkawa, M., 2020. Structural basis for two-way communication between dynein and microtubules. *Nat. Commun.* 11, 1038. <https://doi.org/10.1038/s41467-020-14842-8>

Walton, T., Wu, H., Brown, A., 2021. Structure of a microtubule-bound axonemal dynein. *Nat. Commun.* 12, 477 . <https://doi.org/10.1038/s41467-020-20735-7>

Lacey, S.E., He, S., Scheres, S.H., and Carter, A.P. (2019). Cryo-EM of dynein microtubule-binding domains shows how an axonemal dynein distorts the microtubule. *Elife* 8.

Nishida, N., Komori, Y., Takarada, O., Watanabe, A., Tamura, S., Kubo, S., Shimada, I., and Kikkawa, M. (2020). Structural basis for two-way communication between dynein and microtubules. *Nat Commun* 11, 1038.

Decision Letter, first revision:

29th Jun 2021

Dear Jack,

Thank you for submitting your revised manuscript "Structures of outer-arm dynein array on microtubule doublet reveal a motor coordination mechanism" (NSMB-A44607A). It has now been seen by the original referees and their comments are below. The reviewers find that the paper has improved in revision, and therefore we'll be happy in principle to publish it in *Nature Structural & Molecular Biology*, pending minor revisions to satisfy the referees' final requests and to comply with our editorial and formatting guidelines.

Thank you again for your interest in *Nature Structural & Molecular Biology*. Please do not hesitate to contact me if you have any questions.

Kind regards,
Florian

Florian Ullrich, Ph.D.
Associate Editor
Nature Structural & Molecular Biology
ORCID 0000-0002-1153-2040

Reviewer #1 (Remarks to the Author):

The authors addressed my comments. I fully support the publication.

Reviewer #2 (Remarks to the Author):

The manuscript has been indicated by all three reviewers as a significant contribution to the understanding of dyneins and cilia motility. The revised manuscript did address all the concerns of the reviewers. Therefore I think it is ready to be published by NSMB.

Reviewer #3 (Remarks to the Author):

Revisions based on reviewers's comments more clarified the procedure of experiments and the claims of this paper.

The reviewer thinks this manuscript is now ready to be published in NSMB, but still requires several minor revisions.

(a) Related to Reviewer#3 - Major Comments(1), Table 1 is still confusing.

It is better to add lines to separate the columns in the area below #1 OAD-PF MTBS-1, #2 OAD-PF MTBS-2, and #3 pre-parallel OAD.

(b) Related to Reviewer#3 - Minor Comments(3), no additional information is found in Fig 3a and the legends, in spite of the author's answer: "We have revised the model and explained more details about the model in the figure legend."

Please check the version of the figure 3a and its legend.

Author Rebuttal, first revision:

Reviewer #3 (Remarks to the Author):

Revisions based on reviewers's comments more clarified the procedure of experiments and the claims of this paper.

The reviewer thinks this manuscript is now ready to be published in NSMB, but still requires several minor revisions.

(a) Related to Reviewer#3 - Major Comments(1), Table 1 is still confusing.

It is better to add lines to separate the columns in the area below #1 OAD-PF MTBS-1, #2 OAD-PF MTBS-2, and #3 pre-parallel OAD.

(b) Related to Reviewer#3 - Minor Comments(3), no additional information is found in Fig 3a and the legends, in spite of the author's answer: "We have revised the model and explained more details about the model in the figure legend."

Please check the version of the figure 3a and its legend.

(a) We have added the lines into the Table 1 to separate the columns among #1 OAD-PF MTBS-1, #2 OAD-PF MTBS-2, and #3 pre-parallel OAD.

(b) We apologize that there were indeed some formatting mistakes and appreciate the reviewer's rigorous reading. We have double checked and confirm the figure legends have been updated this time.

Final Decision Letter:

2nd Aug 2021

Dear Jack,

We are now happy to accept your revised paper "Structures of outer-arm dynein array on microtubule doublet reveal a motor coordination mechanism" for publication as a Article in Nature Structural & Molecular Biology.

Before the manuscript is sent to the printers, we shall make any detailed changes in the text that may be necessary either to make it conform with house style or to make it intelligible to a wider readership. If the changes are extensive, we will ask for your approval before the manuscript is laid out for production. Once your manuscript is typeset you will receive a link to your electronic proof via email within 20 working days, with a request to make any corrections within 48 hours. Please read proofs with great care to make sure that the sense has not been altered. If you have queries at any point during the production process then please contact the production team at rjsproduction@springernature.com. Once your paper has been scheduled for online publication, the Nature press office will be in touch to confirm the details.

Please note that due to tight production schedules, proofs should be returned as quickly as possible to avoid delaying publication. If you anticipate any limitations to your availability over the next 2-4 weeks (such as vacation or traveling to conferences, etc.), please e-mail rjsproduction@springernature.com as soon as possible. Please provide specific dates that you will be unavailable and provide detailed contact information for an alternate corresponding author if necessary.

As soon as your article is published, you can generate your shareable link by entering the DOI of your article here: `http://authors.springernature.com/share`. Corresponding authors will also receive an automated email with the shareable link

Your paper will be published online soon after we receive proof corrections and will appear in print in the next available issue. You can find out your date of online publication by contacting the production

team shortly after sending your proof corrections. Content is published online weekly on Mondays and Thursdays, and the embargo is set at 16:00 London time (GMT)/11:00 am US Eastern time (EST) on the day of publication. Now is the time to inform your Public Relations or Press Office about your paper, as they might be interested in promoting its publication. This will allow them time to prepare an accurate and satisfactory press release. Include your manuscript tracking number (NSMB-A44607B) and our journal name, which they will need when they contact our press office.

About one week before your paper is published online, we shall be distributing a press release to news organizations worldwide, which may very well include details of your work. We are happy for your institution or funding agency to prepare its own press release, but it must mention the embargo date and Nature Structural & Molecular Biology. If you or your Press Office have any enquiries in the meantime, please contact press@nature.com.

Please note that *Nature Structural & Molecular Biology* is a Transformative Journal (TJ). Authors may publish their research with us through the traditional subscription access route or make their paper immediately open access through payment of an article-processing charge (APC). Authors will not be required to make a final decision about access to their article until it has been accepted. [Find out more about Transformative Journals](https://www.springernature.com/gp/open-research/transformative-journals)

Authors may need to take specific actions to achieve compliance with funder and institutional open access mandates. For submissions from January 2021, if your research is supported by a funder that requires immediate open access (e.g. according to [Plan S principles](https://www.springernature.com/gp/open-research/plan-s-compliance)) then you should select the gold OA route, and we will direct you to the compliant route where possible. For authors selecting the subscription publication route our standard licensing

terms will need to be accepted, including our [self-archiving policies](https://www.springernature.com/gp/open-research/policies/journal-policies). Those standard licensing terms will supersede any other terms that the author or any third party may assert apply to any version of the manuscript.

Kind regards,
Florian

Florian Ullrich, Ph.D.
Associate Editor
Nature Structural & Molecular Biology
ORCID 0000-0002-1153-2040
